

# Beyond self-healing: Stabilizing and destabilizing photochemical adjustment of the ozone layer

Aaron Match[1], Edwin P. Gerber[1], and Stephan Fueglistaler[2]

[1]Center for Atmosphere Ocean Science, Courant Institute of Mathematical Sciences, New York University
[2]Program in Atmospheric and Oceanic Sciences, and Department of Geosciences, Princeton University

**Correspondence:** Aaron Match (aaron.match@nyu.edu)

**Abstract.** The ozone layer is often noted to exhibit *self-healing*, whereby a process that depletes ozone can nonetheless lead to increased ozone at lower altitudes. Self-healing has been explained to occur because ozone depletion aloft allows more ultraviolet (UV) light to reach lower levels, where it enhances ozone production. Similarly, a process that increases ozone can nonetheless reduce ozone below, known as reverse self-healing. This paper considers self-healing and reverse self-healing to manifest a more general mechanism we call *photochemical adjustment*, whereby ozone perturbations lead to a downward cascade of anomalies in ultraviolet fluxes and ozone. Conventional explanations for self-healing suggest that photochemical adjustment is stabilizing, i.e., the initial perturbation in column ozone is damped towards the surface. However, if the enhanced ultraviolet transmission due to ozone depletion disproportionately increases the ozone sink, then photochemical adjustment can be destabilizing. We use the coefficients of the Cariolle v2.9 linear ozone model to analyze photochemical adjustment in the chemistry-climate model MOBIDIC. We find that: (1) photochemical adjustment is destabilizing in the upper stratosphere, and (2) self-healing is often just the tip of the iceberg of large photochemical stabilization throughout the mid- and lower-stratosphere. The photochemical regimes from MOBIDIC can be reproduced by the Chapman Cycle, a classical model of ozone photochemistry whose simplicity admits theoretical insight. Photochemical regimes in the Chapman Cycle are controlled by the spectral structure of the perturbed ultraviolet fluxes. The transition from photochemical destabilization to stabilization occurs at the slant column ozone threshold where ozone becomes optically saturated in the overlap window of absorption by $O_2$ and $O_3$, i.e., $10^{18}$ molec cm$^{-2}$ (around 40 km in the tropics).

## 1 Introduction

Earth's atmosphere is shaped primarily by just three radiatively-active gases: water vapor, carbon dioxide, and ozone. Water vapor is concentrated near the surface due to temperature-dependent condensation (e.g., Sherwood, 1996; Pierrehumbert and Roca, 1998; Galewsky et al., 2005). Carbon dioxide is well-mixed and evolves slowly subject to surface sources and sinks. Ozone, the focus of this paper, maximizes in the stratosphere where it shields the surface from dangerous ultraviolet radiation. Ozone is formed by photolysis and destroyed by both photolysis and catalytic chemistry.

The photochemical nature of the ozone layer can cause counterintuitive responses to perturbations. It is well known that emitting an ozone-depleting substance whose local chemical effects strictly reduce ozone can nonetheless cause ozone to





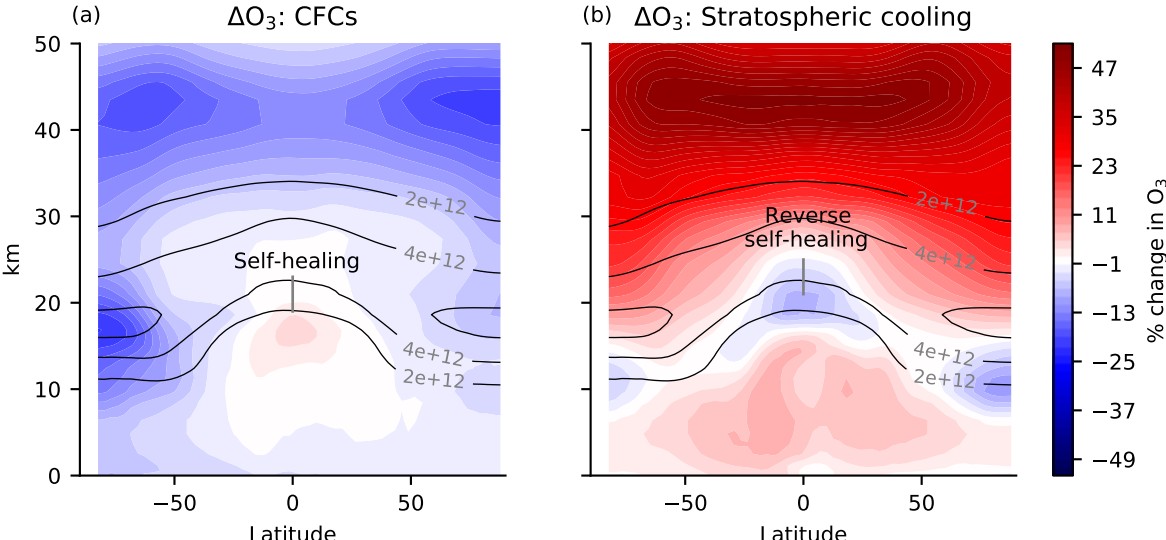

**Figure 1.** Canonical examples of self-healing and reverse self-healing. (a) Percent change in ozone in response to 2014 halocarbons (e.g., CFCs) imposed on a pre-industrial atmosphere (piClim-HC minus piControl) in MRI-ESM2-0, accessed from the CMIP6 archive. Halocarbons generally reduce ozone except in the tropical lower stratosphere, where there is self-healing. Pre-industrial control ozone is contoured in solid lines (molec cm$^{-3}$). (b) The ozone response to stratospheric cooling from a quadrupling of $CO_2$ with pre-industrial SST (piClim-4xCO2 minus piControl) in MRI-ESM2-0, accessed from the CMIP6 archive. Stratospheric cooling generally increases ozone except in the lower stratosphere where there is reverse self-healing.

increase in certain locations (Johnston, 1972; Dütsch, 1979; WMO, 1985; Solomon et al., 1985; Fomichev et al., 2007; Meul et al., 2014). This phenomenon is known as *self-healing*. Self-healing has been explained to occur because the reductions in ozone aloft allow more ultraviolet photons to reach lower altitudes, where they can increase the source of ozone enough to lead to net increases of ozone. Figure 1a provides a canonical example of self-healing in response to ozone-depleting substances. The self-healing effect is visible as increases in ozone in the tropical lower stratosphere (and upper troposphere). Similarly,

perturbations that locally increase ozone can lead to decreases in ozone known as reverse self-healing (Groves et al., 1978; Haigh and Pyle, 1982; Jonsson et al., 2004; SPARC, 2018). Figure 1b provides a canonical example of reverse self-healing in response to stratospheric cooling from the direct radiative effects of elevated $CO_2$. Stratospheric cooling generally increases ozone but leads to reductions in the tropical lower stratosphere.

     Explanations for self-healing and reverse self-healing invoke the fact that photolysis rates depend on attenuation of ultraviolet

fluxes by overhead column ozone. Yet, despite invoking fundamental photochemical principles, self-healing and reverse self-healing typically only arise as curious aspects of the ozone response to perturbations (Figure 1). Self-healing or reverse self-healing have not been studied as objects of direct interest, and they are not described by a quantitative theory. This paper seeks to better understand self-healing and reverse self-healing, and to do so, we seek to understand the full consequences of the





mechanism proposed to drive them. The driving mechanism of self-healing and reverse self-healing in response to a column
ozone perturbation is the downward cascade of perturbation photolysis rates and perturbation column ozone. We define the
component of the ozone response to a perturbation that results from changes in photolysis rates due to changes in overhead
ozone as *photochemical adjustment*. Self-healing and reverse self-healing refer only to the subset of photochemical adjustment
in which the sign of the ozone response to the perturbation is opposite what is expected, but photochemical adjustment is found
to be a more general and fundamental mechanism of ozone photochemistry.

Photochemical adjustment is *stabilizing*, or compensating, if a reduction in column ozone overhead leads to an increase in
ozone below. This increase in ozone will absorb ultraviolet radiation to counteract the effects of the initial reduction in column
ozone, thereby damping the ultraviolet anomaly with altitude towards the surface. Self-healing and reverse self-healing both
result from photochemical stabilization. Photochemical adjustment is *destabilizing*, or amplifying, if a reduction in column
ozone overhead leads to a further reduction of ozone below, thereby amplifying photochemical anomalies towards the surface.
Self-healing and reverse self-healing are typically explained in a way that would seem to preclude the possibility of photochem-
ical destabilization. This is because the net response of ozone is assumed to be determined by how the perturbation ultraviolet
fluxes affect the ozone source. However, in ozone photochemistry, ultraviolet absorption can act as both a source *and* a sink
of ozone. If the ozone sink were more sensitive than the ozone source to perturbations in overhead ozone, then reductions in
ozone aloft that allowed more ultraviolet radiation to lower levels would lead to reductions in ozone below, i.e., photochemical
destabilization.

Self-healing and reverse self-healing have historically been defined to occur where the sign of the ozone response is opposite
that which is expected from the local perturbation. The more general phenomenon of photochemical adjustment cannot be
defined merely by the sign of the ozone response, because photochemical stabilization can mute the ozone response to a
perturbation without changing the sign of the net response, and photochemical destabilization amplifies the response without
changing its sign. Thus, photochemical adjustment must be defined in terms of magnitudes. This paper develops a framework
for defining and quantifying photochemical adjustment through the full ozone profile.

In Section 2, we review current explanations for self-healing and reverse self-healing. Photochemical adjustment is defined,
and we raise the possibility of photochemical destabilization. In Section 3, we quantify photochemical regimes in a chemistry-
climate model with the aid of a pre-existing dataset: the coefficients of a linear ozone model. The atmosphere is found to exhibit
both photochemical stabilization and photochemical destabilization. In Section 4, we propose a quantitative definition of pho-
tochemical adjustment. The coefficients of the linear ozone model are used to develop a photochemical adjustment emulator for
a chemistry-climate model, which reveals that self-healing and reverse self-healing are just the tip of the iceberg of large pho-
tochemical adjustment throughout the entire ozone layer in response to perturbations. In Section 5, photochemical adjustment
in the chemistry-climate model is shown to be strongly predictable from the Chapman Cycle of ozone photochemistry, despite
its omission of key catalytic chemistry and transport. Possible reasons for this good fit are discussed. Based on the Chapman
Cycle, a theory is developed in Section 6 for the transition altitude between photochemical destabilization and photochemical
stabilization. Theoretical predictions of the transition altitude perform well in the chemistry-climate model. In Section 7, we




develop a new qualitative explanation for photochemical adjustment that correctly extends to both photochemical stabilization and photochemical destabilization.

## 2    Qualitative explanations for self-healing and reverse self-healing

An early mention of "self-healing" was in a paper considering the ozone response to emissions of ozone-depleting substances from supersonic transports, when Johnston (1972) wrote, "These profiles include the partial self-healing effect of reformation of ozone at lower elevations." Self-healing or reverse self-healing have subsequently been invoked in hundreds of papers when considering the ozone layer response to CFCs (e.g., NAS, 1976), stratospheric cooling (e.g., Groves et al., 1978), and the evolution of atmospheric oxygen (e.g., conceptually in Kasting and Donahue, 1980). Despite being frequently invoked, self-healing and reverse self-healing have not been explained by a quantitative theory. They are seldom discussed in textbooks (although some qualitative descriptions are available, e.g., Andrews et al., 1987; Finlayson-Pitts and Pitts, 2000). Self-healing and reverse self-healing have mainly been invoked qualitatively and as needed to explain unexpected responses of ozone to perturbations. These invocations share a universal mechanism, yet some explanations also emphasize additional mechanisms.

The universal mechanism among explanations of self-healing is that when ozone is depleted aloft, such as from CFCs, some photons that would have been absorbed by ozone aloft can penetrate more deeply into the atmosphere. These photons can photolyze $O_2$, a reaction that drives the production of ozone. If the enhanced production of ozone leads to a net increase of ozone compared to the unperturbed state, then this is called self-healing. The increased ozone from self-healing partially stabilizes the downwelling ultraviolet flux against the initial depletion of ozone. When ozone is enhanced aloft, such as from stratospheric cooling, the mechanism runs in reverse, leading to reverse self-healing: some photons that would have been absorbed by $O_2$ below are now intercepted by the enhanced $O_3$ aloft. This reduces the ozone source below, and can lead to reductions of ozone, which again partially stabilize the downwelling ultraviolet flux against the initial depletion of ozone. These two mechanisms for photochemical stabilization are shown schematically in the top row of Figure 2.

As is noted in some studies, this universal explanation is premised on the substantial spectral overlap between absorption by $O_3$ and $O_2$ at wavelengths less than 240 nm. For example, Crutzen (1974) wrote: "This self-healing effect is due to deeper penetration of the ozone producing ultraviolet radiation in the 2000-2300 Å[200-230 nm] region." This overlap is necessary because self-healing results when absorption of photons switches from $O_3$ at high altitudes to $O_2$ at low altitudes, and reverse self-healing results when absorption of photons switches from $O_2$ at low altitudes to $O_3$ at high altitudes.

Some explanations for self-healing and reverse self-healing invoke auxiliary mechanisms as well. We remark on two: first, some studies note that self-healing and reverse self-healing are muted by transport (Solomon et al., 1985). This is because self-healing and reverse self-healing tend to occur in the lower stratosphere, where photochemical equilibration times are slow enough that transport can induce substantial deviations in ozone from photochemical equilibrium. Second, some explanations note that the perturbation ultraviolet flux can drive photolysis reactions other than ozone production. When ozone is depleted aloft, the enhanced ultraviolet fluxes can also photolyze ozone, producing atomic oxygen. WMO (1985) noted that this atomic oxygen can then produce hydroxyl radicals that either produce ozone through smog chemistry or destroy ozone in catalytic




cycles. Ozone production through the smog pathway leads to "ozone self-healing of the second kind". Although seldom discussed, this smog pathway might contribute to the increases of ozone in the upper troposphere in Figure 1a. Also seldom discussed but of central interest in this paper is that the perturbation atomic oxygen can also contribute to the destruction of ozone. If the perturbation ultraviolet fluxes can drive ozone destruction, then an underappreciated phenomenon becomes

possible: self-amplified destruction.

Self-amplified destruction can occur if an $O_3$ perturbation aloft leads to perturbation UV fluxes that drive further net destruction of $O_3$. This destruction of $O_3$ can result from the photolysis of $O_3$, which produces O. Much of this O will recombine with $O_2$ to reform $O_3$, completing a null cycle with no net effect on $O_3$, but some of the O can bond with $O_3$ to destroy $O_3$, or can bond with catalysts of $O_3$ depletion, indirectly destroying the $O_3$ by virtue of not closing the null cycle. To the extent that there

is leakage of O from the null cycle into other reactions that destroy $O_3$, photolysis of $O_3$ becomes a sink of $O_3$. Self-amplified destruction is photochemically destabilizing, meaning that the perturbation ultraviolet flux grows with altitude towards the surface. Running this response in reverse, a perturbation that added ozone aloft could reduce the ozone sink below to lead to enhanced ozone below (*self-amplified production*). Figure 2 illustrates these two cases of photochemical destabilization.

Conventional explanations for self-healing and reverse self-healing do not hint at the possibility of photochemical desta-

bilization. Instead, these explanations imply that the net change in ozone is dictated by the changes in the photolytic source alone, without considering possible changes in the photolytic sink. Yet, the prospect of photochemical destabilization can be traced back to Hartmann (1978), who identified significant photochemical destabilization above 40 km in a simple model of the ozone layer. The prospect of photochemical destabilization has been neglected in the intervening decades for two main reasons. The first reason is theoretical: photolysis of ozone has been treated as having no net effect on ozone because of the

significant null cycle, whereas photochemical destabilization requires that photolysis of ozone destroys some amount of ozone. The second reason is practical: self-healing and reverse self-healing demand an explanation because the sign of the ozone response is unexpected, whereas photochemical destabilization demands no such explanation because it does not change the sign of the ozone response. Photochemical destabilization can only be distinguished by the *magnitude* of the response, and thus, in the absence of quantitative treatments, photochemical destabilization would be aliased into the local response of the

ozone layer to the initial perturbation. De-aliasing photochemical destabilization begins by quantifying the sensitivity of ozone photochemistry to perturbations in column ozone, as calculated in a chemistry-climate model in the next section.

## 3    Photochemical regimes in a chemistry-climate model

An ideal test of whether the ozone layer is photochemically stabilizing or destabilizing in a given location is to calculate the sensitivity of the local net production rate of ozone to an ultraviolet perturbation induced by some change in the overlying

column ozone, holding all else fixed. Increasing the overlying column ozone will decrease the incoming photons in accord with the spectrally-varying absorption of ozone and the magnitude of the perturbation. This perturbation to the incoming photons can modify both ozone production and destruction, so the key response variable is the change in net production rate of ozone (i.e., production minus loss). If an increase in the overlying column ozone reduces the net production rate, then that location





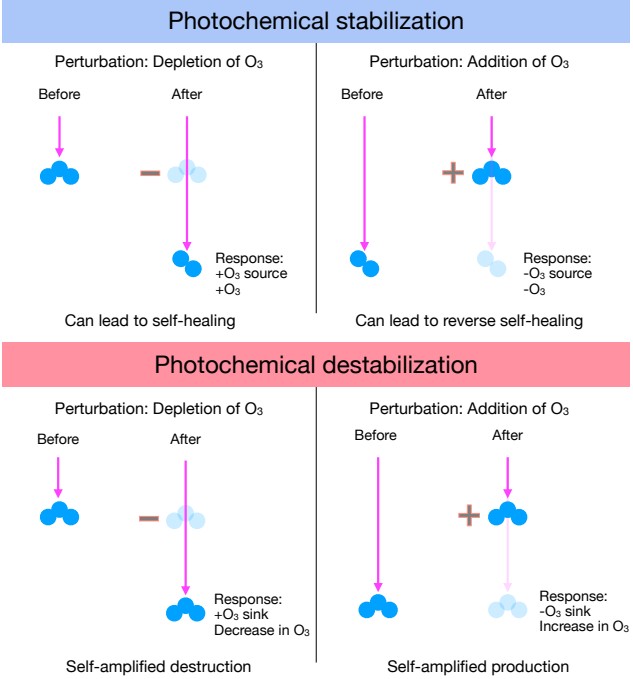

**Figure 2.** Photochemical stabilization is typically explained with the mechanisms in the top row. (Top left) Depletion of ozone aloft allows deeper penetration of UV that can increase the ozone source and produce additional ozone, potentially leading to self-healing. (Top right) Addition of ozone aloft prevents deeper penetration of UV, decreasing the ozone source below and removing ozone, potentially leading to reverse self-healing. Photochemical destabilization is proposed to result from the mechanisms in the bottom row. (Bottom left) Depletion of ozone aloft allows deeper penetration of UV that can increase the ozone sink leading to self-amplified destruction. (Bottom right) Addition of ozone aloft prevents deeper penetration of UV, decreasing the ozone sink below and increasing ozone, leading to self-amplified production.

is in a photochemically stabilizing regime. If an increase in the overlying column ozone increases the net production rate, then
that location is in a photochemically destabilizing regime. To characterize a climatology of photochemical regimes, this test would have to be repeated at all latitudes, altitudes, and seasons.

This exact battery of regime tests has been performed in chemistry-climate models by previous studies that developed linear ozone models. Linear ozone models are tools for computationally speeding up chemistry-climate modeling of ozone. In a linear ozone model, the net tendencies of ozone from a chemistry-climate model with 10s-100s of reactions are linearized with
respect to finite perturbations in local ozone concentration, temperature, and column ozone. Then, for computational efficiency, the ozone module can be replaced with these linear functions evaluated using their table of coefficients. A standard linear ozone model equation, first proposed in Cariolle and Déqué (1986), is as follows:

$$\frac{dr_{O_3}}{dt} = A_1 + A_2(r_{O_3} - A_3) + A_4(T - A_5) + A_6(\chi_{O_3} - A_7) \tag{1}$$



where $r_{O_3}$ is the ozone mixing ratio, $A_1 = P - L$ is the basic state photochemical production minus loss rate, $A_2 = \partial(P - L)/\partial r_{O_3}$, $A_3$ is the basic state ozone mixing ratio, $A_4 = \partial(P-L)/\partial T$, $A_5$ is the basic state temperature, $A_6 = \partial(P-L)/\partial \chi_{O_3}$ where $\chi_{O_3}$ is the overlying column ozone $\int_z^\infty [O_3]dz$ in units of [molec cm$^{-2}$] and [O$_3$] is the ozone number density [molec cm$^{-3}$], and $A_7$ is the basic state column ozone.

We propose to interpret the linear ozone model coefficients as classifying the photochemical regime based on the sign of the sensitivity to perturbations in column ozone ($A_6$). Photochemical stabilization occurs where $A_6 < 0$, meaning that an increase in overhead ozone reduces the net ozone production. Photochemical destabilization occurs where $A_6 > 0$, meaning that an increase in overhead ozone increases the net ozone production.

Previous studies have calculated the $A$ parameters within a chemistry-climate model as functions of altitude, latitude, and month. We have analyzed the $A_6$ coefficients in the Cariolle v2.9 linear ozone model. We thank Daniel Cariolle for sharing the linear ozone model with us. The Cariolle v2.9 linear ozone model is descended from the first linear ozone model, with a major update to include heterogeneous chemistry (not directly relevant to this work) in Cariolle and Teyssèdre (2007). The Cariolle linear ozone model is based on the chemistry-climate model MOBIDIC, first described in Cariolle and Brard (1985). The transport is 2D based on the residual circulation. The recent version of MOBIDIC to which the Cariolle v2.9 linear ozone model is calibrated includes the main gas-phase reactions driving the NOx, HOx, ClOx, and BrOx catalytic cycles, with 30 transported long-lived species and 30 short-lived species computed diagnostically (Cariolle and Teyssèdre, 2007). The Cariolle linear ozone model has been shown to perform well compared to state-of-the-art comprehensive chemistry-climate models (Meraner et al., 2020) and is used to prognose ozone in the ERA5 reanalysis (Hersbach et al., 2020).

The values of $A_6$ during the month of March (chosen as an equinoctial month) are shown in Figure 3a. Negative values indicate photochemical stabilization, and positive values indicate photochemical destabilization. A major result is that there is significant and large photochemical destabilization above 40 km. This means that ozone depletion above 40 km removes additional ozone down to 40 km. Photochemical destabilization occurs above the maximum number densities of O$_3$; as evaluated using the climatological structure of O$_3$ in MOBIDIC, 4% of ozone molecules in the deep tropics (from 20°S-20°N) are in a photochemically destabilizing regime above 40 km.

Intriguingly, the photochemical destabilization above 40 km in the tropics is located where it was first noted in Hartmann (1978) for the Chapman Cycle model. The Chapman Cycle model only involves four chemical reactions—two photolysis reactions and two combination reactions—yet has been shown to capture many gross features of the ozone layer, such as the overall shape of the ozone layer (Chapman, 1930; Jacob, 1999). Figure 3b shows sensitivity of the net ozone production rate for a Chapman photochemical equilibrium to perturbations in column ozone. This Chapman Cycle result has been placed beside the Cariolle v2.9 result to emphasize that the Chapman Cycle reproduces the map of the photochemical regimes. The details of the Chapman Cycle result and the theory (magenta curves) will be presented later in Section 5.

The coefficients of the Cariolle v2.9 linear ozone model have provided exactly the evidence needed to classify regions of the atmosphere as photochemically stabilizing or destabilizing. One striking feature of Figure 3a is that the sensitivity of net ozone production rate to perturbations in column ozone has a larger magnitude in the destabilizing region than in the stabilizing region. This might seem to imply that destabilization is stronger than stabilization. However, the basic state production and loss are also





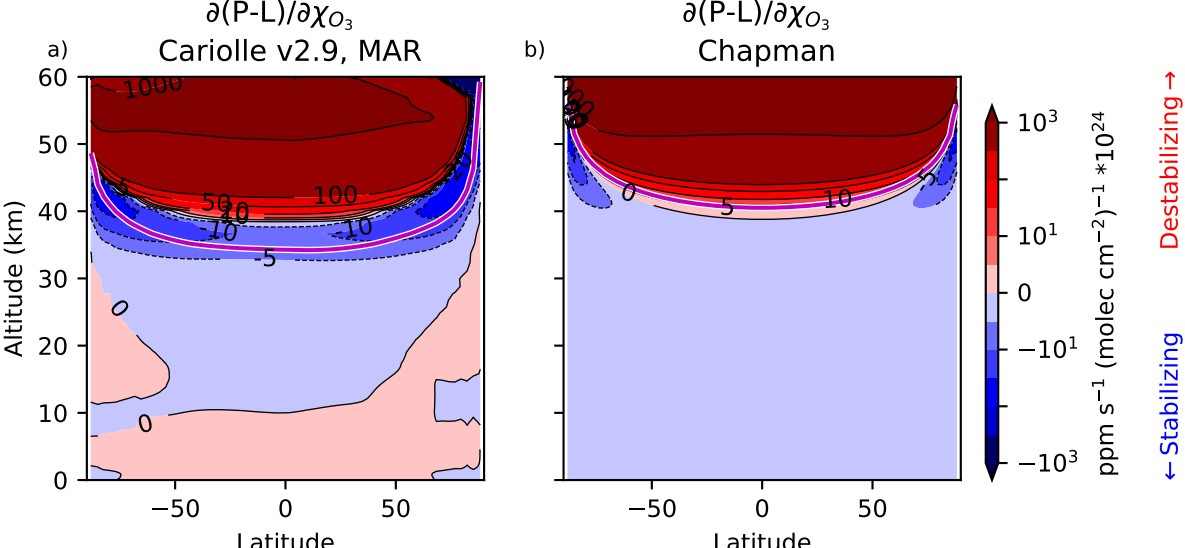

**Figure 3.** Sensitivity of net ozone photochemical production rate to perturbations in column ozone in (a) Cariolle v2.9 in March (approximately equinoctial conditions such that latitude $\theta \approx$ solar zenith angle) and (b) the Chapman model, in which latitudinal dependence is represented by performing single-column calculations. Photochemical stabilization is indicated by negative values, whereas photochemical destabilization is indicated by positive values. Spectrally-resolved photochemical theory (Section 6) predicts positive coefficients where the slant ozone column is less than $10^{18}$ molec cm$^{-2}$ (i.e., above the magenta curve).

larger in the upper atmosphere, where the actinic flux has not been attenuated as strongly, so therefore the ozone equilibration
timescale may also be relevant to considering the magnitude of photochemical stabilization. Questions about the magnitude of photochemical stabilization demand a method for quantifying the full vertical profile of photochemical adjustment in general, a method we develop in the following section.

## 4 Defining photochemical adjustment and quantifying it in the linear ozone model

Photochemical adjustment is the component of the ozone response to a perturbation that results from changes in photolysis due
to changes in column ozone aloft. This is the idea behind self-healing—the surprising increase in ozone in the lower tropical stratosphere in simulations of ozone depletion is attributed to the increase in photons penetrating to lower levels—but to date it lacks a quantitative assessment. We therefore formally define the photochemical adjustment to an ozone perturbation as the difference between a fully interactive simulation of ozone photochemistry and one where the UV-flux (and hence photons available for photolysis of $O_2$ and $O_3$) is held fixed:

Photochemical adjustment $\equiv [O_3]|_{\text{Fully Interactive}} - [O_3]|_{\text{UV-Locked}}$ (2)





This UV-locking method can in principle be implemented in any modeling context for atmospheric chemistry. Intuitively, a UV-locked simulation can be obtained in a chemistry climate model by fixing the overlying column ozone seen when calculating photolysis rates at its unperturbed value. This ensures that the photolysis rates do not respond to any changes in ozone resulting from the perturbation, but allows all other adjustments to the chemistry.

In this paper, we will implement this approach using two methods. First, we use the linear ozone model coefficients to develop a linear emulator for analyzing ozone perturbations in MOBIDIC. Later, we will perform various UV-locking experiments in the Chapman Cycle, which will be shown to reproduce essential aspects of the chemistry-climate model.

We begin our derivation for the linear emulator of the perturbation ozone equation by converting ozone anomalies in the linear ozone model from mixing ratio (Equation 1) into number density by multiplying by the number density of air, denoted

$n_a$ [molec cm$^{-3}$]. We then decompose the variables in our prognostic ozone number density equation into basic state and perturbation components, $x = \bar{x} + x'$ , where $\bar{x}$ is the basic state component and $x'$ the perturbation. Subtracting the basic state component from the total equation, and assuming a quasi-steady state allows us to solve for the perturbation ozone number density, as a function height, in the presence of some prescribed perturbation, $G$:

$$[\mathrm{O}_3]' = -\frac{A_4 n_a T'}{A_2} - \frac{A_6 n_a \chi'_{\mathrm{O}_3}}{A_2} + G \tag{3}$$

On the right-hand-side, the first term captures the sensitivity to temperature perturbations, the second term captures the sensitivity to column ozone perturbations, thereby encoding the photochemical adjustment, and the third term is the prescribed perturbation in ozone. $A_2$ (defined as $\partial(P-L)/\partial r_{\mathrm{O}_3}$) tends to be negative, reflecting that local perturbations in ozone tend to decay on a timescale of $-1/A_2$. While Equation 3 is linear at any given height, it is mathematically implicit because the perturbation column ozone is the integral of the ozone number density perturbation (i.e., $\chi'_{\mathrm{O}_3} = \int_z^\infty [\mathrm{O}_3]' dz$). Photochemical

adjustment at height thus influences lower levels, introducing a spatial nonlinearity encoded in the $A_6$ term, and one must solve the system downward from the top of the atmosphere.

We use the linear ozone model to quantify the response of the tropical ozone layer to ozone depletion and to stratospheric cooling. Ozone depletion is represented as a uniform reduction of ozone below its basic state value ($G = -G_0$, $T' = 0$), which in turn can induce photochemical perturbations by perturbing the actinic flux that drives ozone at lower levels. Thus, the fully

interactive linear response to this uniform perturbation in ozone includes the perturbation itself plus the nonlocal effects of perturbed actinic flux. Stratospheric cooling is represented as a uniform cooling (G = 0, $T' = -\Delta T$), which tends to locally increase ozone. These local increases in ozone can then perturb the actinic flux, leading to a fully interactive response that includes the local effects of cooling and the nonlocal effects of column ozone changes aloft.

These two perturbations are explored by using the coefficients of the linear ozone model to perform an offline calculation

of the linear response of a steady state ozone layer to ozone depletion or stratospheric cooling. The linear ozone model distills the relative importance of the local effects of our perturbations from the non-local effects of the actinic flux perturbations from aloft, as conveyed by the sensitivity to column ozone. Both of these offline calculations have been performed on the discrete grid of the linear ozone model, with levels $z_i$, i = 1-N numbered from the top of the atmosphere downwards, each





with a depth of $\Delta z_i$. Uniform ozone depletion with a magnitude $G_0$ has been imposed at and below $z_5 = 60$ km. The fully
interactive ozone response to this perturbation has been calculated by iterating between calculating the ozone response at
level $i$ and the resulting column ozone anomaly at level $i$ that is then seen by level $i + 1$. The initial anomaly is therefore
$[O_3]'(z_5) = -G_0$, which leads to a column ozone anomaly of $\chi'_{O_3}(z_5) = -G_0 \Delta z_5$. The fully interactive response at $z_6$ can then
be calculated as $[O_3]'(z_6) = -G_0 - A_6 n_a \chi'_{O_3}(z_5)/A_2$. This ozone anomaly is used to update the column ozone perturbation as
$\chi'_{O_3}(z_6) = \chi'_{O_3}(z_5) + [O_3]'(z_6)\Delta z_6$. The solution can then proceed iteratively, where at each lower altitude, we first calculate
the ozone perturbation, then add that ozone perturbation (multiplied by the layer depth) to the column ozone perturbation.

A similar iterative method is used to calculate the response to stratospheric cooling. Here, we consider a uniform cooling
of $\Delta T$ throughout the atmosphere, a notable simplification on the actual stratospheric cooling in response to greenhouse gases
given that we impose such cooling throughout the stratosphere and troposphere and with no vertical structure. Our iterative
calculation begins at the top of the atmosphere with a local ozone anomaly of $[O_3]'(z_1) = A_4 n_a \Delta T/A_2$. This local ozone
anomaly leads to a column ozone perturbation of $\chi'_{O_3}(z_1) = [O_3]'(z_1)\Delta z_1$. The ozone perturbation at the next lower altitude
can then be calculated in general as $[O_3]'(z_i) = A_4 n_a \Delta T/A_2 - A_6 n_a \chi'_{O_3}(z_{i-1})/A_2$. The column ozone perturbation can then
be updated in general as $\chi'_{O_3}(z_i) = \chi'_{O_3}(z_{i-1}) + [O_3]'(z_i)\Delta z_i$.

Using this approach, we calculated the linear response of ozone to uniform ozone depletion and stratospheric cooling. The
response to these perturbations in the absence of photochemical adjustment can also be calculated directly as the UV-locked
response by repeating the calculation with $\chi'_{O_3}$ overwritten to zero everywhere. For uniform ozone depletion, the UV-locked
response is simply the prescribed ozone depletion, a reduction by $G_0$ at all altitudes. For stratospheric cooling, the UV-locked
response is the direct thermal response given by the first term on the right-hand-side of Equation 3. Knowing the photolysis-
locked responses, it is then straightforward to calculate the photochemical adjustment using Equation 2 as the Fully Interactive
response minus the UV-Locked response.

The results are shown in Figure 4, as calculated using the linear ozone model coefficients in March for an equatorial atmo-
spheric column. The response to uniform ozone depletion of $G_0 = 10^8$ molec cm$^{-3}$ is shown in Figure 4a, with the prescribed
depletion shown in blue. The fully interactive response (black) differs substantially from the prescribed perturbation. The fully
interactive response shows reductions of ozone in the upper stratosphere that are larger in magnitude than those prescribed,
peaking at 42 km with an amplitude 75% larger than what is prescribed. This is a consequence of self-amplified destruction in
the region of photochemical destabilization above 40 km. Below 40 km, the response is no longer photochemically destabiliz-
ing, and the fully interactive response is more positive than the prescribed reduction. At 35 km, the fully interactive response
becomes positive in absolute terms; this is self-healing, an increase in ozone driven by a perturbation that reduces ozone lo-
cally. The fully interactive increase in ozone peaks at $2G_0$ at 29 km; the associated photochemical adjustment here is $3G_0$. The
photochemical adjustment decays towards the lower stratosphere and troposphere, where the ozone response eventually resem-
bles the prescribed depletion. The overall column response of ozone is substantially modified by photochemical adjustment,
as can be seen by integrating these curves. The column integral of the prescribed perturbation is $-G_0 * 60$ km. The column
integral of the fully interactive response is only half as large, with the difference due to photochemical adjustment. Overall,
photochemistry significantly compensates for the initial perturbation.





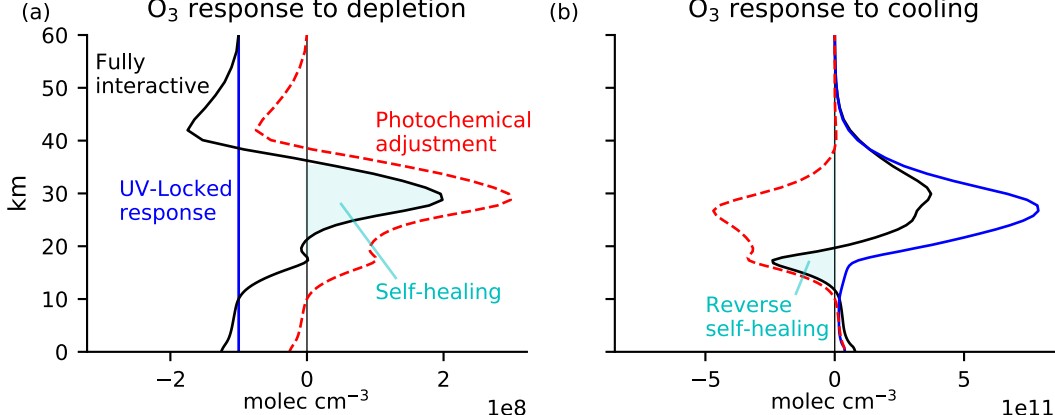

**Figure 4.** Ozone response to (a) uniform depletion by $10^8$ molec cm$^{-3}$ everywhere below 60 km and (b) uniform cooling of 10 K. The calculation is performed in a linear emulator of MOBIDIC by iteratively solving Equation 3 using coefficients from the Cariolle v2.9 linear ozone model. The Fully Interactive response (black) is decomposed into a UV-Locked response (blue) and photochemical adjustment (red). Self-healing and reverse self-healing (cyan shading) are merely the tip of the iceberg of large photochemical adjustment throughout the column, with photochemical destabilization above 40 km and photochemical stabilization below 40 km.

The response to uniform stratospheric cooling of $\Delta T = 10$ K is shown in Figure 4b. The UV-Locked response (blue)
is a strict increase in ozone. Cooling approximately rescales ozone by a constant factor, so the ozone response is shaped approximately like the underlying ozone layer. The fully interactive response (black) differs substantially from the UV-locked response, especially in the lower stratosphere. The maximum increase in ozone is predicted in both cases to occur around 30 km, but the Fully Interactive response is only half that predicted under UV-locking. This discrepancy is due to photochemical adjustment, which is large and stabilizing throughout much of the stratosphere. Below 20 km, ozone is reduced in absolute
terms (reverse self-healing). Integrating the column responses, the increase in total column ozone in the Fully Interactive case is only 43% as large as the increase in UV-locked; again, photochemical adjustment has halved the perturbation compared to in its absence.

A major result of this analysis is that photochemical adjustment is large and significant throughout the entire stratosphere. It is important here to distinguish between photochemical adjustment and self-healing/reverse self-healing. We can express the
colloquial definition for self-healing or reverse self-healing as follows:

$$\text{(Reverse) self-healing} \equiv \begin{cases} [\text{O}_3]'_{\text{Fully Interactive}}, & \text{if sign}([\text{O}_3]'_{\text{Fully Interactive}}) * \text{sign}([\text{O}_3]'_{\text{UV-locked}}) < 0 \\ 0, & \text{otherwise} \end{cases}$$

This definition identifies that self-healing or reverse self-healing are given by the ozone perturbation in a fully interactive simulation conditional on that perturbation having the opposite sign to what is expected. Expressed this way, it is clear that





self-healing and reverse self-healing are not the same as photochemical adjustment (defined in Equation 2). Indeed, self-healing
and reverse self-healing are often just the tip of the iceberg of much larger photochemical stabilization. This is especially clear
in the response to stratospheric cooling (Figure 4b), in which the column-integrated reverse self-healing is only 14% as large
as the column-integrated photochemical stabilization.

The two main results of our analysis using the Cariolle v2.9 linear ozone model are that (1) photochemical adjustment
is destabilizing above 40 km and (2) photochemical adjustment can be large throughout the stratosphere, much greater than
implied by conventional self-healing and reverse self-healing. Photochemical destabilization is inconsistent with the standard
explanation for self-healing and reverse self-healing, and therefore demands a revision to the conceptual understanding of these
ubiquitous phenomena. This new conceptual understanding will also provide quantitative insights.

## 5    Understanding photochemical adjustment with the Chapman Cycle

The large photochemical destabilization above 40 km and the large photochemical adjustment throughout the stratosphere (not
just in regions of self-healing or reverse self-healing) both demand a revision to present understanding. New understanding
can be challenging to achieve in a comprehensive chemistry-climate model with tens of prognostic chemical species under-
going hundreds of reactions. Yet, some basic properties of photochemical adjustment do not depend on this complexity, and
are reproducible from simpler models. In particular, the prospect of photochemical destabilization was first raised in Hartmann
(1978), who found such behavior above 40 km in a modified version of the Chapman Cycle. It is at precisely those altitudes that
MOBIDIC also simulates photochemical destabilization, suggesting that the Chapman Cycle might hold promise for under-
standing photochemical adjustment despite omitting many chemical processes that are known to be important. Jumping ahead
to results, Figure 3 showed that the Chapman Cycle (panel b) reproduces the photochemical regimes from a comprehensive
chemistry-climate model (panel a).

### 5.1    Methods: The Chapman Cycle model

The Chapman Cycle is the textbook model of the ozone layer that explains its gross features, e.g., why the ozone layer is in the
stratosphere (e.g., Jacob, 1999; Brasseur and Solomon, 2005). The Chapman Cycle considers the photochemical equilibrium
obtained when ultraviolet radiation impinges on an atmosphere with oxygen species undergoing the following reactions:

$$O_2 + h\nu \quad \rightarrow \quad O + O \qquad (\lambda < 240\,\text{nm}) \tag{R1}$$

$$O + O_2 + M \quad \rightarrow \quad O_3 + M \tag{R2}$$

$$O_3 + h\nu \quad \rightarrow \quad O_2 + O \qquad (\lambda < 320\,\text{nm}) \tag{R3}$$

$$O + O_3 \quad \rightarrow \quad O_2 \tag{R4}$$

where Reactions R1 and R3 are photolysis reactions and $M$ is a third body. We will now assume that the molar fraction of
$O_2$ is constant, because it is much larger than the molar fractions of $O_3$ or O. Under this assumption, these reactions can be





manipulated algebraically to yield a quadratic equation for ozone at a given altitude as it depends on the ultraviolet fluxes and
collisional reaction rates (Craig, 1965):

$$[O_3] = \frac{k_2}{k_4} C_{O_2} n_a^2 \frac{k_1 C_{O_2} n_a}{k_1 C_{O_2} n_a + k_3[O_3]} \tag{4}$$

where the rates of each reaction are indicated $k_i$ (i=1-4), $C_{O_2}$ is the molar fraction of $O_2$ (0.21), and $n_a$ is the number density
of air (molec cm$^{-3}$). Following a similar derivation, the atomic oxygen can be expressed as a diagnostic function of ozone
(Craig, 1965):

$$[O] = \frac{k_1[O_2] + k_3[O_3]}{k_2 C_{O_2} n_a^2} \tag{5}$$

These equations can be simplified for stratospheric conditions. Through much of the stratosphere, the majority of photons
are absorbed by ozone compared to $O_2$, i.e., $k_3[O_3] \gg k_1 C_{O_2} n_a$, leading Equation 4 to simplify to the following standard
approximation:

$$[O_3] = \left( \frac{k_1 k_2}{k_3 k_4} \right)^{1/2} C_{O_2} n_a^{3/2} \tag{6}$$

Ozone appears implicitly on the right-hand side of Equation 6 through the photolysis rates $k_1$ (s$^{-1}$) and $k_3$ (s$^{-1}$) which are
governed by the spectrally-integrated ultraviolet absorption:

$$k_1(z) = \int_\lambda q_{O_2}(\lambda) \sigma_{O_2}(\lambda) I_\lambda(z) d\lambda \tag{7}$$

$$k_3(z) = \int_\lambda q_{O_3}(\lambda) \sigma_{O_3}(\lambda) I_\lambda(z) d\lambda \tag{8}$$

with wavelength $\lambda$, quantum yield $q_i(\lambda)$ (molecules decomposed per photon absorbed by species $i$), absorption coefficient $\sigma_i$
(cm$^2$ molec$^{-1}$) shown in Figure 5, and actinic flux density with respect to wavelength $I_\lambda(z)$ (photons cm$^{-2}$ s$^{-1}$ nm$^{-1}$). The
absorption of photons by photolysis attenuates the actinic flux:

$$I_\lambda(z) = I_\lambda(\infty) \exp\left( -\frac{\tau_\lambda(z)}{\cos\theta} \right) \tag{9}$$

where $I_\lambda(\infty)$ is the incoming actinic flux, $\theta$ is the solar zenith angle, and $\tau_\lambda$ is the optical depth as a function of wavelength.
The optical depth at a given altitude depends on column-integrated $O_2$ and $O_3$ above that level:

$$\tau_\lambda(z) = \sigma_{O_2}(\lambda) \chi_{O_2}(z) + \sigma_{O_3}(\lambda) \chi_{O_3}(z) \tag{10}$$

where $\chi_{O_2} = \int_z^\infty [O_2] dz$ and $\chi_{O_3} = \int_z^\infty [O_3] dz$.



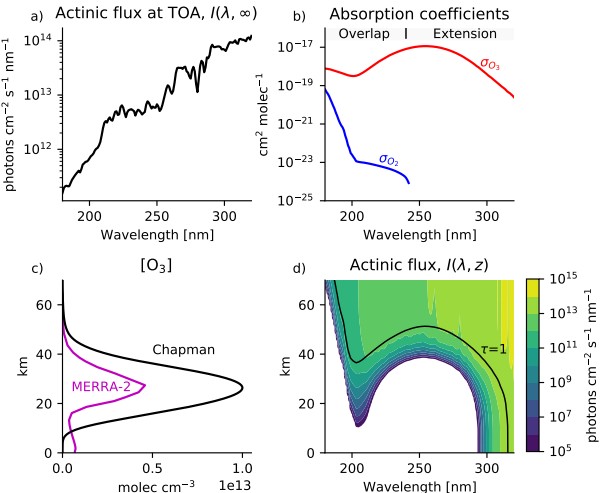

**Figure 5.** Basic state of the Chapman model. (a) Actinic flux at the top of the atmosphere. (b) Absorption coefficients for $O_2$ and $O_3$. (c) Chapman $O_3$ profile (black) versus MERRA-2 $O_3$ profile from 30°S-30°N in 2018 (magenta). (d) Actinic flux in Chapman model resolved in wavelength and altitude.

### 5.1.1 Chapman Cycle numerics

We use a modified version of the Chapman implementation described in Match and Gerber (2022) in photochemical equilibrium mode with an isothermal atmosphere and a default of overhead sun ($\cos \theta = 1$). We neglect transport, tropospheric chemistry,
scattering, clouds, and surface reflection. Equation 4 is solved at each altitude downward from the top of the atmosphere to calculate the photochemical equilibrium profile (Craig, 1965; Ratner and Walker, 1972).

The vertical dimension is discretized into vertical levels ($\Delta z$ = 100 meters) ranging from the surface to 100 km. The idealized shortwave radiative transfer and photolysis rates are solved on a wavelength grid with 141 discretized wavelengths ranging from 180 nm to 320 nm. Spectrally-resolved parameters are linearly interpolated to the wavelength grid. Solar actinic flux is
calculated from the Solar Spectral Irradiance Climate Data Record (Coddington et al., 2015), averaged from 01-01-2020 to 02-04-2021. $O_2$ absorption coefficients ($\sigma_{O_2}$) are taken from Ackerman (1971) and $O_3$ absorption coefficients ($\sigma_{O_3}$) from Demore et al. (1997). The isothermal atmosphere has a default temperature of 240 K and scale height of 7 km. Temperature-dependent parameters for reaction rates $k_2(T)$ and $k_4(T)$ are taken from Brasseur and Solomon (2005).

Chapman photochemical equilibrium is calculated numerically. The parameters and resulting photochemical equilibrium are
shown in Figure 5. Figure 5c shows the Chapman Cycle photochemical equilibrium profile of ozone (black) compared to the tropical-averaged profile in MERRA-2 for the representative year of 2018 (magenta). Chapman photochemical equilibrium is well known to overestimate ozone by approximately a factor of two because it neglects transport and catalytic ozone destruction (Bates and Nicolet, 1950; Crutzen, 1970; Jacob, 1999; Brasseur and Solomon, 2005). Figure 5d shows the actinic flux as a




function of wavelength and altitude in the photochemical equilibrium solution, with a contour indicating the $\tau = 1$ level where the actinic flux has been attenuated by a factor of $e$ compared to the top-of-atmosphere actinic flux.

## 5.2 Photochemical regimes in the Chapman model

Conceptually, the photochemical regimes in the Chapman Cycle are controlled by the relative sensitivity of photolysis of $O_2$ and photolysis of $O_3$ to perturbations in column ozone. In response to perturbations in column ozone, photochemical stabilization occurs where the photolytic source of ozone is more sensitive than the photolytic sink; destabilization occurs where the photolytic sink of ozone is more sensitive than the photolytic source. Of course, photolysis of $O_3$ acts as a sink of ozone with an efficiency inversely related to the strength of the null cycle R3 $\rightleftharpoons$ R2. A quantitative calculation of the photochemical sensitivity of the Chapman Cycle must take this null cycle into account.

We calculate the photochemical sensitivity by quantifying how the net production rate of ozone (source minus sink) changes in response to a perturbation in column ozone while holding ozone itself locally fixed. (Holding ozone fixed is essential, because if ozone was allowed to adjust, then photochemical equilibrium would prevail and the source would necessarily balance the sink in both the control and perturbed states.) The net production rate of ozone in the Chapman Cycle comes from R2, R3, and R4:

$$P - L = k_2[O][O_2][M] - k_3[O_3] - k_4[O][O_3] \tag{11}$$

Substituting in the expression for atomic oxygen (Equation 5) leads to the following expansion of the net production rate after grouping terms by their dependence on $k_1$ versus $k_3$:

$$P - L = k_1[O_2](1 - \epsilon) - \epsilon k_3[O_3] \tag{12}$$

where $\epsilon \equiv k_4[O_3]/k_2 C_{O_2} n_a^2$ indicates the destruction efficiency with which photolysis of ozone leads to the destruction of ozone, and can be seen to equal the the rate of R4 (by which atomic oxygen destroys ozone) divided by the rate of R2 (by which atomic oxygen produces ozone). Photolysis of ozone ($k_3$) evidently acts as a loss of ozone because it appears exclusively in a negative term. However, this sink of ozone is not one-to-one with the photolysis events, because much photolysis of $O_3$ drives the null cycle from R3 to R2. At 35 km, $\epsilon \sim O(10^{-3})$, meaning that out of 1000 photolysis events of ozone, 999 will proceed into the null cycle and only 1 will actually turn the odd oxygen back into $O_2$. The strength of the null cycle (1 - $\epsilon$) also modulates the effectiveness of photolysis of $O_2$ at producing ozone. If the null cycle is weak such that atomic oxygen tends to destroy ozone, then photolysis of $O_2$ would not be as strong a source of ozone. The smallness of $\epsilon$ (i.e., $\epsilon < 1$) guarantees that photolysis of $O_2$ is always a source of ozone.

Holding ozone fixed at its control value of $\overline{[O_3]}$, we can differentiate Equation 12 with respect to column ozone, where the only terms that depend on column ozone are the photolysis rates $k_1$ and $k_3$:



$$\frac{\partial(P-L)}{\partial\chi_{O_3}} = \frac{\partial k_1}{\partial\chi_{O_3}}[O_2](1-\epsilon) - \epsilon\frac{\partial k_3}{\partial\chi_{O_3}}\overline{[O_3]} \tag{13}$$

The ozone layer is photochemically stabilizing when $\frac{\partial(P-L)}{\partial\chi_{O_3}} < 0$ and photochemically destabilizing when $\frac{\partial(P-L)}{\partial\chi_{O_3}} > 0$. The

destruction efficiency $\epsilon$ can be calculated from the basic state solution of ozone and actinic fluxes. It turns out that the partial derivatives of the photolysis rates to perturbations in column ozone can also be calculated explicitly by noting that photolysis rates depend on column ozone through the term $\exp\left(-\sigma_{O_3}(\lambda)\chi_{O_3}/\cos\theta\right)$, which can be differentiated with respect to $\chi_{O_3}$ to yield a factor of $-\sigma_{O_3}(\lambda)/\cos\theta$. The resulting sensitivities (i.e., the partial derivatives of Equations 7 and 8 with respect to column ozone) are as follows:

$$\frac{\partial k_1}{\partial\chi_{O_3}} = -\int_\lambda q_{O_2}(\lambda)\sigma_{O_2}(\lambda)\overline{I_\lambda}(z)(\frac{\sigma_{O_3}(\lambda)}{\cos\theta})d\lambda \tag{14}$$

$$\frac{\partial k_3}{\partial\chi_{O_3}} = -\int_\lambda q_{O_3}(\lambda)\sigma_{O_3}(\lambda)\overline{I_\lambda}(z)(\frac{\sigma_{O_3}(\lambda)}{\cos\theta})d\lambda \tag{15}$$

where $\overline{I_\lambda}(z)$ is the basic state actinic flux. Equations 14 and 15 are strictly negative, confirming that increasing column ozone can only reduce photolysis rates by attenuating the actinic flux. Evaluating Equations 14 and 15 only requires variables diagnosed from the basic state (e.g., the basic state actinic flux) and integrals with respect to wavelength. Importantly, Equations

14 and 15 are not mathematically implicit (i.e., perturbation ozone does not appear implicitly on the right-hand-side) and do not require an iterative integral in altitude. Thus, Equations 14 and 15 can be substituted into Equation 13 to yield an explicit formula for the sensitivity of the ozone net production rate to perturbations in column ozone in terms of only local properties of the ozone layer and basic state variables.

This explicitly-calculated photochemical sensitivity of the Chapman Cycle is shown in Figure 3b. The solar zenith angle

has been approximated as the latitude to allow a comparison with the photochemical sensitivity of MOBIDIC during the equinoctial month of March (Figure 3a). The Chapman Cycle reproduces the gross features of MOBIDIC, most importantly the destabilizing regime above 40 km in the tropics and the stabilizing regime below. The Chapman Cycle reproduces the latitudinal dependence in which the transition between the destabilizing regime and the stabilizing regime shifts up to higher altitudes at higher latitudes.

Interpreting Equation 13, it is evident that the conventional explanation for self-healing, which assumes that photolysis is a source of ozone, neglects the sensitivity of the ozone sink ($\partial k_3/\partial\chi_{O_3}$). The full Chapman Cycle analysis reveals that the photochemical sensitivity of the ozone layer is controlled by the relative magnitudes of the sensitivity of $k_1$ and $k_3$ (the latter being weighted by the destruction efficiency). Equations 14 and 15 indicate that the sensitivities of the photolysis rates are controlled by the covariance between the spectrally-resolved photolysis rate in the basic state and the absorption coefficient of

ozone. This presages that the spectral structure of the ozone absorption coefficients will critically control the photochemical





sensitivity, a theme that we will return to in Section 6. For now, we proceed to a calculation of the full profile of photochemical adjustment in the Chapman Cycle.

### 5.3 Photochemical adjustment in the Chapman Cycle

As with MOBIDIC, we calculate the photochemical adjustment of the Chapman Cycle in response to ozone depletion and
cooling. It is not trivial to drive a vertically uniform loss of ozone in a fully interactive system, even one this simple, due to the vertical coupling through the actinic flux. The effect of ozone depletion is therefore constructed with an iterative calculation of ozone and column ozone, working from the top of the atmosphere downwards. Starting at the uppermost level, we first compute the equilibrium $[O_3]$, and then overwrite it with our ozone loss perturbation, $[O_3]$-$G_0$. At the next level down, we can then compute the equilibrium $[O_3]$, where the photolysis rates depend on the perturbed ozone column above. Again, this
is overwritten with the imposed loss: $[O_3]$-$G_0$, and the calculation proceeds iteratively downwards yielding an ozone profile with a uniform deficit of ozone, but otherwise consistent with the actinic flux. Stratospheric cooling is comparatively more straightforward to implement, incorporated through the temperature dependence of the collisional rates $k_2$ and $k_4$, using the functions for temperature dependence from Brasseur and Solomon (2005).

The UV-locked experiment is performed by locking both $k_1$ and $k_3$ to their control profiles. More refined photolysis-locking
is also possible in the Chapman Cycle, because we can separately lock $k_1$ or $k_3$. This allows us to test the prevailing explanation for the response of the ozone layer to perturbations, which neglects the sensitivity of the ozone sink ($k_3$) to column ozone perturbations. We perform Lock Sink experiments in which $k_3$ is locked, quantitatively enforcing this common assumption to test whether it reproduces the Fully Interactive response.

The results for quantifying photochemical adjustment in the Chapman Cycle in response to ozone depletion and cooling
are shown in Figure 6. These results reproduce the key structure of those from the linear emulator of MOBIDIC in Figure 4, including our two major results: (1) photochemical adjustment is destabilizing above 40 km and (2) photochemical adjustment is large throughout the stratosphere, with self-healing and reverse self-healing often just the tip of the iceberg.

The Lock Sink experiments provide a unique result from the Chapman Cycle experiments, which could not be performed in the linear emulator of MOBIDIC (because it did not distinguish between the ozone source and sink). If the prevailing
explanation for photochemical adjustment applies to the Chapman Cycle, then photochemical adjustment would be dominated by changes in the ozone source with minimal contributions from changes in the ozone sink. Thus, the prevailing explanation for photochemical adjustment can be quantitatively tested by comparing the Lock Sink experiment (magenta) against the Fully Interactive experiment (black). This comparison reveals major limitations in the prevailing explanation. By construction, the Lock Sink experiments cannot produce photochemical destabilization, because photochemical destabilization occurs when the
sink is more sensitive than the source. Because it cannot produce photochemical destabilization, the Lock Sink experiment fails to predict the large self-amplified destruction above 40 km in response to ozone depletion. Around 45 km in the Fully Interactive experiment, the contribution from self-amplified destruction actually exceeds the local magnitude of $G_0$, leading to absolute reductions of ozone exceeding $2G_0$. At these levels of maximum self-amplified destruction, the Lock Sink experiment



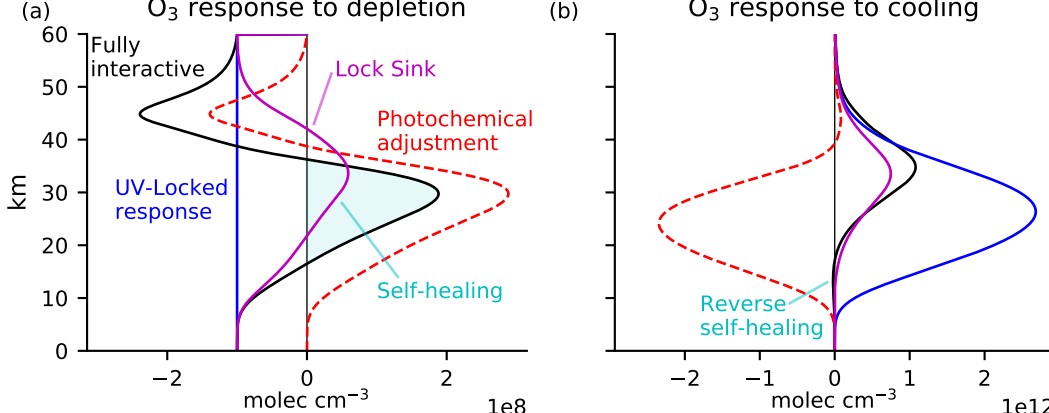

**Figure 6.** The Chapman Cycle response to the same perturbations as in Figure 4, i.e. (a) ozone depletion by $10^8$ molec cm$^{-3}$ everywhere below 60 km and (b) uniform cooling of 10 K. The Chapman Cycle reproduces the response structure calculated using the linear emulator of MOBIDIC. The Chapman Cycle also permits the calculation of a Lock Sink response (magenta), for which only $k_3$ was locked while $k_1$ was allowed to adjust to the overhead ozone, emulating the conventional explanation for self-healing and reverse self-healing, yet failing to reproduce key aspects of the ozone response to perturbations.

wrongly predicts photochemical stabilization. The substantial self-healing is therefore biased towards higher altitudes in the
Lock Sink experiment compared to in the Fully Interactive experiment.

In the Cooling experiment, the response under UV-Locking is a rescaling of the ozone layer by a constant factor. (In the linear emulator of MOBIDIC, the response is not strictly a rescaling because the background state is not isothermal, ozone is not in photochemical equilibrium, and other processes beyond the Chapman Cycle are represented; the overall response to cooling has a similar shape to the Chapman Cycle solution but smaller magnitude.) In both the Chapman Cycle and the
linear emulator of MOBIDIC, the *Fully Interactive* response of the ozone layer quickly deviates from the direct temperature response due to photochemical adjustment (red dashed curve). Photochemical adjustment generally stabilizes the ozone layer in response to cooling, although there is subtle destabilization, again above 40 km. Reverse self-healing, in which ozone is reduced in *Fully Interactive* compared to *Control* (negative values of the black curve), is generally considered the primary manifestation of photochemical stabilization. Yet, reverse self-healing only captures a sliver of the much-larger photochemical
stabilization (negative values of the red curve). Reverse self-healing is stronger in the Off-Line Cariolle v2.9 emulator than in the Chapman Cycle, suggesting a role for non-Chapman stabilizing processes including catalytic chemistry and transport (Wuebbles et al., 1983; Solomon et al., 1985; WMO, 1985).





# 6 A spectral interpretation of photochemical stabilization and destabilization

Ozone depletion and cooling caused both photochemical destabilization and photochemical stabilization, with the Chapman

Cycle and linear emulator of MOBIDIC both simulating a transition from destabilization aloft to stabilization below around 40 km in the tropics. Analysis of the Chapman Cycle reveals that this regime transition is controlled by which parts of the absorption spectrum for $O_3$ and $O_2$ have been attenuated versus which parts of the absorption spectrum are still active. Recall that absorption of photons by $O_2$ drives the production of $O_3$, whereas absorption of photons by $O_3$ drives the destruction of $O_3$. Two key spectral windows will turn out to determine the overall photochemical regime: (1) the *overlap window*, which

has absorption by both $O_2$ and $O_3$, occurring for $\lambda < 240$ nm, and (2) the *extension window*, which has absorption by only $O_3$, occurring for $\lambda > 240$ nm. These absorption spectra are shown in Figure 5b.

Photochemical stabilization can only occur in the overlap window where photons can switch between being absorbed by $O_2$ and $O_3$. The overlap window can also support photochemical destabilization. The extension window can only support photochemical destabilization. The relative importance of absorption in these two spectral windows turns out to be dominantly

controlled by a single variable endogenous to the ozone dynamics: column ozone. Under the control of column ozone, the photochemical regimes emerge as follows:

- **Photochemically neutral regime**: When column ozone is so small that the ozone layer is optically thin at all wavelengths, absorption in both windows varies linearly with column ozone, and the ozone layer is photochemically neutral.

- **Photochemically destabilizing regime**: As column ozone increases, the first wavelength to saturate with respect to

absorption by ozone (i.e., exceed $\tau_{O_3} \approx 1$) occurs at the peak absorption coefficients of ozone of $\sigma_{O_3} \approx 10^{-17}$ cm$^2$ molec$^{-1}$. Unit optical depth is thus first reached for a column ozone of $\chi_{O_3} = 1/\sigma_{O_3,\text{max}} \approx 10^{17}$ molec cm$^{-2}$. The wavelengths where absorption first saturates are at the center of the Hartley band, which is in the extension window. Therefore, at these column ozone values, absorption in the extension window is more sensitive to perturbations in column ozone than absorption in the overlap window, which remains unsaturated. Because absorption in the extension window is

by ozone only, this means that the ozone sink is more sensitive to perturbations in column ozone than the ozone source, a destabilizing regime.

- **Photochemically stabilizing regime**: As column ozone increases further, the overlap window saturates around a column ozone of $10^{18}$ molec cm$^{-2}$. Beyond this column ozone threshold, the ozone source in the overlap window attenuates exponentially as a function of column ozone, whereas the ozone sink in the extension window still has some unsaturated

wavelengths that attenuate linearly as a function of column ozone. Thus, absorption in the overlap window becomes more sensitive than in the extension window, stabilizing the ozone layer.

The destabilization layer robustly terminates at 40 km in response to diverse perturbations because that is where column ozone surpasses the stabilization threshold of $10^{18}$ molec cm$^{-2}$. These column ozone thresholds have been validated with mechanism denial experiments using the Chapman Cycle. For example, we flattened the absorption coefficients in the overlap

window and then artificially varied their magnitudes across three orders of magnitude. As expected, destabilization initiates





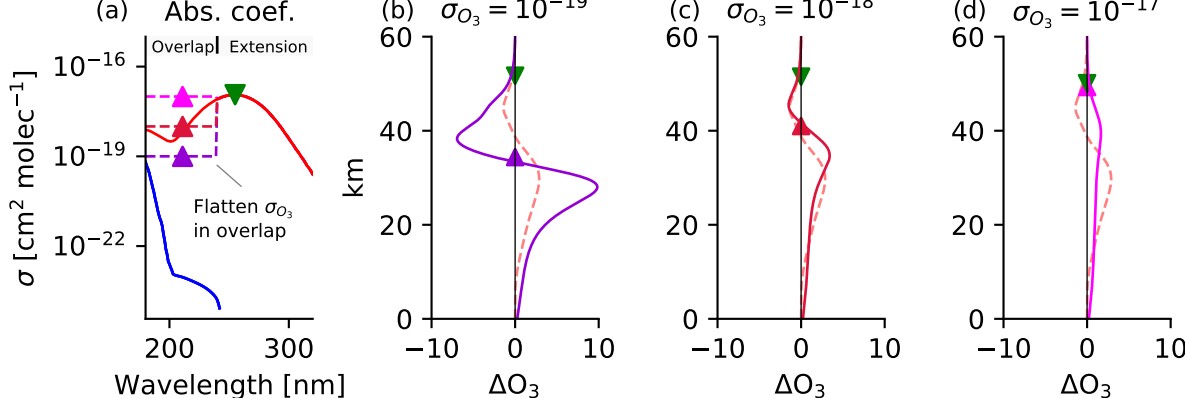

**Figure 7.** Photochemical adjustment in the Chapman model in response to uniform ozone depletion under different constant values of $\sigma_{O_3}$ in the overlap window. (a) The absorption spectra where $\sigma_{O_3}$ in the overlap window is flattened at various constant values. (b-d) For each constant values of $\sigma_{O_3}$, photochemical adjustment is calculated in response to uniform ozone depletion of $G_0 = 10^8$ molec cm$^{-3}$ below 60 km, normalized by dividing by $G_0$. Spectral theory (Section 6) correctly predicts that destabilization ($\Delta O_3 < 0$) begins at the column ozone threshold where the extension window begins to saturate (green downward-pointing triangle). Spectral theory also correctly predicts the transition from destabilization to stabilization at the column ozone threshold where the overlap window saturates (upward-pointing triangles), which varies across the three experiments. Photochemical adjustment using the true absorption coefficients (as seen in Figure 6a) is plotted as the red dashed curve (identical in panels b-d).

once the peak absorption of the Hartley band saturates, and stabilization initiates once the overlap window saturates. Figure 7 shows these results.

These column ozone thresholds have been validated for the linear ozone model coefficients calculated from MOBIDIC. Figure 3 includes magenta curves corresponding to the slant column ozone threshold of $10^{18}$ molec cm$^{-2}$, where slant column ozone has been calculated as $\chi_{O_3}/\cos(\theta)$ with $\theta$ taken as the latitude (noting that we chose an equinoctial month to approximately equate latitude with zenith angle, although neglecting the diurnal cycle in sun angle). In MOBIDIC, the climatological column ozone is given by parameter $A_7$, and in the Chapman Cycle, the climatological ozone is calculated as the photochemical equilibrium profile at each latitude, approximating $\theta$ with latitude. Although not perfect, the theoretical curve approximately reproduces the transition from destabilization to stabilization, including its latitudinal structure (Figure 3). At high latitudes, the transition from destabilization to stabilization occurs at a higher altitude due to the oblique angle of the sunlight, which reaches the slant column ozone threshold more rapidly for a given ozone profile.





## 7 Updating the explanation for self-healing and reverse self-healing

Photochemical models produce photochemical adjustment as a natural consequence of accurately representing photochemistry. Thus, the key results of this paper are primarily conceptual, since they inform how the sensitivity of the ozone layer to pertur-
bations is interpreted and explained. These results could also have implications for certain decisions made when formulating photochemical models, considered in the Discussion. However, the main conceptual advances from this paper suggest that some of the previous explanations of ozone self-healing and reverse self-healing could lead readers to develop a distorted view of photochemical adjustment.

Most explanations suggest that self-healing and reverse self-healing can be understood by considering the effect of the per-
turbed ultraviolet fluxes on the photolytic ozone source. For example, self-healing is explained in Hudson (1977) as follows: "$O_3$ destruction at the upper levels in the stratosphere is partially compensated by the increase in $O_3$ at the lower levels due to deeper penetration of solar UV radiation." Similarly, in one of the few textbook explanations of self-healing, Finlayson-Pitts and Pitts (2000) wrote: "When stratospheric $O_3$ is removed... there is more light available for photolysis of O2 and the formation of more ozone." Although not strictly wrong, these explanations might lead to the incorrect impression that deeper
penetration of solar UV radiation generally leads to increased ozone. Instead, we have shown that this common explanation applies only to the stabilizing regime of ozone photochemistry, but that there is also a destabilizing regime of ozone photochem-
istry. Photochemical adjustment cannot generally be understood by only considering the effects of the perturbation ultraviolet flux on the ozone source. Neglecting the effects of the perturbation ultraviolet flux on the ozone sink was shown to lead to errors in the magnitude and vertical structure of the photochemical adjustment (Figure 6).

Thus, it seems plausible to formulate a minimal explanation for self-healing and reverse self-healing that is consistent with the reality of photochemical adjustment with reference to a basic state in photochemical equilibrium. Thus, we explain ozone self-healing and reverse self-healing as follows:

- **Self-healing:** Ozone depletion enhances ultraviolet fluxes penetrating to lower altitudes, increasing both the ozone source and the ozone sink. Ozone can exhibit a local net increase (self-healing) if the photolytic ozone source increases more than the photolytic ozone sink and that increase is large enough to overcome any local ozone depletion.

- **Reverse self-healing:** Ozone enhancement reduces ultraviolet fluxes penetrating to lower altitudes, reducing both the ozone source and the ozone sink. Ozone can exhibit a local net decrease (reverse self-healing) if the photolytic ozone source decreases more than the photolytic ozone sink and that decrease is large enough to overcome any local ozone enhancement.

## 8 Discussion

Photochemical adjustment does not involve feedback, i.e., perturbations in column ozone affect ozone below a given altitude, but they do not affect ozone at that altitude. The effects of photochemical adjustment cascade downwards through the col-
umn. Absent a local feedback, photochemical destabilization cannot lead to a local photochemical instability in which ozone



anomalies run away to become arbitrarily large or small. Photochemical adjustment can be analogized to a snowball rolling
down a hill, which grows in proportion to itself in regions of destabilization and shrinks in proportion to itself in regions of
stabilization. This analogy provides intuition to two key aspects of photochemical adjustment: (1) there is no local feedback
(the snowball cannot grow while remaining stationary), and (2) the peak photochemical anomalies are set by the magnitude of
destabilization and the depth of the atmosphere (length of the hill).

## 8.1 The role of transport

The Chapman Cycle was successful at reproducing the photochemical regimes of MOBIDIC, in particular the transition from
destabilization above 40 km to stabilization below 40 km in the tropics. However, below about 25 km in the tropics, transport
of ozone is known to become a leading-order contributor to the ozone budget, and thus the Chapman Cycle assumption of
photochemical equilibrium breaks down. Self-healing and reverse self-healing tend to occur lower in the stratosphere, often
in this transport-dominated region. It has been argued that self-healing and reverse self-healing are damped by transport in
these regions (Solomon et al., 1985). The damping effects of transport on self-healing are important because, by opposing
photochemical stabilization, transport can increase ultraviolet flux perturbations at the surface.

However, it appears that the overall effects of transport on self-healing and reverse self-healing might be nontrivial and
potentially ambiguous. This is because transport opens up a non-equilibrium mechanism for photochemical stabilization: ozone
depletion aloft can allow more ultraviolet fluxes below, speeding up photochemical equilibration in the lower stratosphere
and shifting down the transition altitude between the transport-dominated regime and the photochemical-dominated regime.
This transport-mediated mechanism occurs out of photochemical equilibrium and operates in addition to the photochemical
equilibrium mechanisms considered in this paper. This transport pathway would seem to exclusively lead to photochemical
stabilization, because ozone equilibrates more rapidly under stronger ultraviolet flux. The linear ozone model neglects this
dependence of the ozone equilibration rate ($A_2$) on column ozone. Thus, compared to a motionless atmosphere, an atmosphere
with transport might have an additional mechanism for self-healing and reverse self-healing through changes in the transition
altitude between the transport-dominated regime and the photochemically-dominated regime. Because the effects of transport
seem ambiguous, further work is needed to clarify whether it damps or amplifies self-healing and reverse self-healing.

## 8.2 Implications

Comprehensive photochemical models naturally produce realistic self-healing and reverse self-healing. Thus, the key advances
of this paper are conceptual. They help analysts understand what mechanisms lead to robust model output. The interpretive
advances of this work are most obvious when considering that a linear ozone model already contained regions of photochem-
ical destabilization in the upper stratosphere that had not been described. The concept of photochemical stabilization and
destabilization accommodates major features of chemistry-climate model simulations that might have coexisted uneasily with
previous theory. Nonetheless, this work also has methodological and physical implications that extend beyond understanding
pre-existing model functions.





A methodological implication of this work can be found in the construction of linear ozone models. In McCormack et al. (2006), a linear ozone model is formulated based on the model CHEM2D. In calculating the coefficients for the sensitivity to column ozone perturbations, they encoded into their method the assumption that photolysis of ozone does not contribute to the destruction of ozone. Thus, they calibrated their sensitivities to perturbations in column ozone based only on the effects of changes in the photolytic source of ozone. This approach appears to forbid photochemical destabilization by construction. Their approach is equivalent to our Lock Sink calculation in Figure 6, which was shown to induce large biases compared to a Fully Interactive calculation in the Chapman Cycle. It is possible that these biases could be smaller for a comprehensive calculation, but they do not appear to be *a priori* negligible. The evidence of strong photochemical destabilization in the Cariolle v2.9 model (Figure 3a) and the apparent success of the Chapman Cycle at reproducing that structure (Figure 3b) suggests that the Chapman sink cannot be discounted *a priori* when considering the photochemical sensitivity of the ozone layer.

A physical implication of this work is that, because photochemical destabilization occurs below a threshold of column ozone of roughly $10^{18}$ molec cm$^{-2}$, atmospheres with less ozone could have deeper regions of photochemical destabilization. In today's tropical atmosphere, destabilization occurs above 40 km, encompassing approximately 4% of the ozone layer. This column ozone threshold would occur at a much lower altitude and could encompass a much larger fraction of the ozone layer in an atmosphere with less total ozone. The atmosphere had lower ozone throughout much of Earth's history, over which atmospheric $O_2$ has increased by seven orders of magnitude, and column ozone has correspondingly increased from being vanishingly small to its present values (Kasting and Donahue, 1980). If ozone increased continuously, then there must have been an era when total column ozone was approximately $10^{18}$ molec cm$^{-2}$, during which our theory predicts that the ozone layer would have been maximally photochemically destabilizing all the way to the surface. This could have amplified the effects of processes that add or deplete ozone, thereby amplifying the variability of UV light reaching the surface, affecting early life. Similarly, planets outside our solar system could host ozone layers that are photochemically destabilizing. Ozone layers at varying concentrations of $O_2$ have been simulated using one-dimensional models (Kasting and Donahue, 1980; Levine, 1977; Segura et al., 2003) and recently chemistry-climate models (Cooke et al., 2022; Józefiak et al., 2023), but the photochemical sensitivity of the ozone layer has not been quantified for atmospheres with different amounts of $O_2$, a subject for future work.

## 9 Conclusions

The photochemical nature of the ozone layer has long been known to cause it to respond in surprising ways to perturbations. Ozone-depleting substances can lead to increases in ozone, known as self-healing, which have been argued to result from the increased ultraviolet fluxes reaching lower altitudes. Processes that increase ozone, such as stratospheric cooling, can lead to decreases in ozone, known as reverse self-healing, which have been argued to result from the decreased ultraviolet fluxes reaching lower altitudes. We have pursued a quantitative characterization and understanding of the photochemical processes that lead to self-healing and reverse self-healing. To do so, we have defined photochemical adjustment as the component of the ozone response to a perturbation that results from changes in photolysis due to changes in column ozone aloft. Photochemical



adjustment can be quantified as the difference in the ozone response to a perturbation in a fully interactive photochemical experiment versus in a photochemical experiment with ultraviolet fluxes locked to their control profile.

Photochemical adjustment was quantified using a linear emulator of the chemistry-climate model MOBIDIC based on the coefficients of the Cariolle v2.9 linear ozone model. We studied the response to ozone depletion and stratospheric cooling. Photochemical adjustment can be stabilizing, damping column ozone perturbations towards the surface, and potentially leading to self-healing and reverse self-healing. Self-healing and reverse self-healing only occur for the subset of photochemical stabilization in which the photochemical adjustment dominates the local effects of the perturbation. Thus, self-healing and

reverse self-healing are often just the tip of the iceberg of large photochemical adjustment. We have also characterized a significant region of photochemical destabilization, above 40 km. Photochemical destabilization is unconventional, given that photochemical adjustment was previously only considered in the context of photochemical stabilization leading to self-healing and reverse self-healing.

The regimes of photochemical stabilization and destabilization in MOBIDIC were successfully emulated using the funda-

mental Chapman Cycle model of the ozone layer. The Chapman Cycle reproduces the full latitudinal structure of the transition from destabilization to stabilization (Figure 3). Photochemical destabilization in the Chapman Cycle arises when the photolytic sink of ozone (absorption by $O_3$) is more sensitive than the photolytic source (absorption by $O_2$) to perturbations in overhead $O_3$. The sensitivity of the photolytic source and sink can in turn be related directly to the absorption spectra of $O_2$ and $O_3$. Photochemical destabilization occurs when absorption in the extension window ($\lambda > 240$ nm) is becoming optically saturated

($\tau_{O_3} = 1$) while absorption in the overlap window ($\lambda < 240$ nm) remains unsaturated. Photochemical stabilization occurs once absorption by $O_3$ in the overlap window saturates. This spectral intuition leads to a simple theory that photochemical destabilization should occur below a threshold of slant column ozone of approximately $10^{18}$ molec cm$^{-2}$, which reproduces the latitudinal structure of the transition altitude between destabilization and stabilization in MOBIDIC and the Chapman Cycle.

Roughly 4% of ozone molecules in the tropics today are in a photochemically destabilizing regime. As considered in the

Discussion as a subject for future work, an atmosphere with less total ozone will have a higher fraction of its ozone layer in a destabilizing regime below the column ozone threshold of $10^{18}$ molec cm$^{-2}$. Over Earth's history, atmospheric $O_2$ has increased by at least seven orders of magnitude, leading to the formation of the ozone layer (e.g., Kasting and Donahue, 1980). Paleoclimatic ozone layers with total column ozone below the destabilization threshold could have been destabilizing all the way down to the surface, amplifying ozone perturbations and increasing the variability of UV light experienced by early life.

*Code and data availability.* The Chapman Cycle Photochemical Equilibrium Solver described in Section 5 is published at doi:10.5281/zenodo.10515738. CMIP6 data is accessible from https://esgf-node.llnl.gov/search/cmip6/.

*Author contributions.* Authors' contributions: AM and EPG acquired funding; AM, EPG, and SF conceptualized research; AM performed formal analysis; AM wrote original draft; EPG and SF reviewed and edited paper.



*Competing interests.* The authors declare that they have no conflict of interest.

*Acknowledgements.* The authors thank Daniel Cariolle for sharing the latest version (v2.9) of his linear ozone model. A.M. acknowledges constructive discussions with Benjamin Schaffer, Nadir Jeevanjee, and at the Princeton Center for Theoretical Science workshop *From Spectroscopy to Climate*. This work was supported by the National Science Foundation under Award No. 2120717 and OAC-2004572, and by Schmidt Futures, a philanthropic initiative founded by Eric and Wendy Schmidt, as part of the Virtual Earth System Research Institute (VESRI). For the CMIP6 model output, we acknowledge the World Climate Research Programme, the climate modeling groups, and the

Earth System Grid Federation (ESGF), as supported by multiple funding agencies.



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
