# Peer review of "Beyond self-healing: Stabilizing and destabilizing photochemical adjustment of the ozone layer"

_EGUsphere, 2024_

## Referee Comment (RC2)

*Review of*

**"Beyond self-healing: Stabilizing and destabilizing photochemical adjustment of the ozone layer "**

*by A. Match et al.*

**General**

I like the general idea of the paper, namely a focus on the issue of "self-healing" and to look into the detailed chemical mechanisms involved. And these mechanisms are indeed discussed in detail.

But I also have reservations about the study. First, I think that the paper is not well formulated in many respects; I suggest a "methods" section where the employed models are better described and introduced. And the available literature at this point is rather old. Second, the paper sounds almost like a global study (e.g. abstract), but throughout the paper the tropics are mentioned. (But not the mod-latitudes). So is this a paper on tropical ozone? The neglect of transport seems to limit the range of applicability. Third, I do not agree that the Chapman cycle constitutes a good approximation of the state of the ozone layer; the loss term of ozone in the atmosphere is at no altitude dominated by the Chapman reaction R4 (Fig. 2). Finally, the assumptions on stratospheric cooling and ozone depletion are likely to crude (no time period given, no altitude dependence assumed) so that the question arises what can be learned about the real atmosphere based on such assumptions.

I summary, I think the paper needs work and improvement. The assumptions (and range of applicability) should be clear and there should be a clear message.

**Comments**

**Neglecting stratospheric transport of ozone**

Clearly stratospheric ozone is strongly influenced by transport (Fig. 1 of this review); this is acknowledged by the authors of the paper. However, in the main body of the paper this is only noted in passing (and not at all in the introduction)

[Figure]

**Figure 1.1.2-1** Logarithm of the computed lifetime of the odd oxygen family in Northern Hemisphere winter versus latitude and height, from the Garcia-Solomon two-dimensional model (from Garcia and Solomon, 1985). Regions dominated by chemical and dynamical processes are indicated.

Figure 1: Chemical and dynamical control of ozone in the stratosphere. (from WMO, 1990).

and not really employed in the text. I have noted that there is section 8.1. But in my opinion, the study needs to be clear about in which regions (due to transport) the employed analysis is not applicable. At the moment the study reads like a "photochemistry only" study, where transport is inserted as an 'add on'.

**Chapman cycle**

The paper makes the point that the "Chapman model" explains the gross features of the ozone layer. I am sorry, but I cannot agree. The "Chapman model" is known to be incorrect since decades. At no altitude, the Chapman reaction R4 dominates ozone loss (Fig. 2 of this review; see also Portmann et al. (2012) and the textbooks cited in the manuscript). The total column ozone (referred to in this paper) globally (and seasonally resolved) would look very different than observed, if there was only the "Chapman model". The problem is already visible in Fig. 5C (of the paper), but how would look a comparison of ozone "(Chapman vs. observations)" in mid-latitudes in winter? Or in the polar regions.

[Figure]

Figure 2: The left panel shows the relative global mean ozone loss rates by chemical family computed for 2000 levels of source gases by the NOCAR two-dimensional model. The right panel shows the global mean ozone profile, which highlights the ozone layer maximum in the middle stratosphere. (from Portmann et al., 2012).

The authors have a good point in emphasising the wave length resolved variation of penetration of radiation throughout the stratosphere.

I understand the idea of the study is to quantify the response of the (tropical?) ozone layer to ozone depletion aloft, but in my understanding this requires to get the ozone loss terms (and their dependency on photolysis) right – the "Chapman model" cannot do this.

Finally, I cannot see a conceptional advantage of using the "Chapman model" – the catalytic cycles that impact ozone loss essentially speed up reaction R4 (compared to a pure "Chapman model") so why is it not possible to go this route?

**Range of validity of the results**

The paper should be clear about the rage of validity of the results presented here. If I read the abstract (and the title) it looks like a global analysis. But it is not. Clearly, heterogeneous chemistry (and thus polar ozone loss) is neglected (which would be available in the linear scheme Cariolle and Teyssèdre, 2007), this is

okay. But nonetheless, Fig. 1 extends to the poles.

Clearly tropospheric ozone chemistry is not the point here, this paper is on stratospheric processes. And certainly there is no 10 K cooling in the troposphere. In spite of this, Figs. 1, 6 and 7 extend to the ground.

Moreover, I am not convinced that the processes discussed here are valid for the entire stratosphere. Sometimes, the tropics are mentioned (e.g., Fig. 5c is only for the tropics) and particular effects are only discussed for the tropics (e.g., line 456). Does this mean that the analysis presented here is only for the tropics? If yes it should be stated. And how are the tropics defined here? If both the tropics and the extra-tropics are discussed here, there are likely different processes relevant for these altitudes – do you disagree?

Finally, in the tropics, the mean reaction rate between $\approx$ 20-40 km increases by more than an order of magnitude – I think this process is relevant for the arguments on chemical processes put forward here and should not be ignored.

**The employed assumptions on ozone loss and stratospheric cooling**

This study uses assumptions on ozone loss and stratospheric cooling. These assumptions are supposed to be somewhat realistic to be useful and applicable to the real atmosphere. First, suggesting a change in ozone or temperature without giving a time period is not very useful. If you are looking for numbers of changes that are realistic for particular time periods, WMO (2022) might be helpful. The strongest ozone change is at the poles (not treated here) and in the upper stratosphere.

Second, assuming a uniform (altitude independent) change of ozone throughout the atmosphere is not realistic (WMO, 2022) and this assumption is even worse for temperature (sorry, but you seem to assume 10 degree *reduction* of temperature at ground level ...).

**Ozone ($O_3$), O and the $O_x$ family**

The analysis in the submitted manuscript does not involve a discussion of the $O_x = O_3 + O$ family. The textbook knowledge (conventional wisdom) states that it does not matter whether O or $O_3$ is lost through chemistry, but only if a member of the family $O_x = O_3 + O$ is lost. That is, reaction R4 (in the manuscript) is an $O_x$ loss,

but not R3. But in contrast, R3 is stated in the manuscript as a *loss of ozone*. The reason for using the concept of an $O_x$ family is of course that reactions R2 and R3 are very fast (at least by more than an order of magnitude) compared to all other reactions of importance in the stratosphere. This point seems to be worked out again at the bottom of page 15 in the manuscript, i.e. the null cycle from R3 to R2.

It is of course okay to disagree with the textbook knowledge on the $O_x$ family but I do not recommend to ignore it. If you do something else here I suggest justifying it against the $O_x$ family concept. As I read the paper, R3 is considered as a *loss* of ozone – I do not think this is true, with consequences for the arguments put forward in the manuscript. But I am happy to be convinced otherwise.

However, I think that all the other catalytic cycles impacting stratospheric ozone (and their dependence of the actinic flux at particular wavelengths) should not be ignored.

**Model description and documentation**

The paper uses the MOBIDIC model; I would not call this model a "chemistry-climate model" – it is a two-dimensional model; ("chemistry-climate model" sounds a bit like CMIP, which is misleading). As far as I understand the paper, the chemistry scheme used in MOBIDIC is the Cariolle scheme (Eq. 1) – so I suggest to make this very clear in the paper. (Or the other way around, if I am not correct here). Further, the cited documentation on MOBIDIC is decades old (1985) and what is cited is not an extensive description of the model (but rather a conference contribution). (Is there a more recent description of the model?) It is also not clear from the paper, why for such two dimensional model calculations one needs to use the linear Cariolle scheme, rather than performing a full chemistry simulation. Regarding two-dimensional models, there were other options (e.g. Fleming et al., 2011, 2015, but of course there are other alternatives); I do not think that computational issues are an argument today (for such studies). Did the authors consider testing the performance of the Cariolle scheme against a full chemistry simulation?

I was indeed shown (Meraner et al., cited in the paper) that the annual mean total ozone column and the tropical ozone profile of MOBIDIC agree well for linear and the explicit chemistry schemes. However, other issues remain. For the purpose of the present study, I think that in particular the partial ozone column *above* some altitude is important. Further, is the performance outside of the tropics relevant here? If not, this needs to be explicitly discussed. If I understand correctly,

MOBIDIC has issues with $HNO_3$, especially in high latitudes in summer – is this correct?

Also, MOBIDIC and the Cariolle scheme have certainly evolved over time during the decades since the cited reference in 1985. I would be good to have at least some information on the new parameters of the Cariolle scheme (see also below). Further, in the paper there is a discussion of a "fully interactive" calculation (see also below); there is some description of the calculation in the paper, but I suggest to make it very clear right from the start, which calculations and which models are used here (and which are not used, no tropospheric chemistry, no heterogeneous chemistry). Perhaps there could be a methods section in the paper, where such things are discussed?

**Abstract and title**

ACP suggests that titles should be concise and consistent with the content and purpose of the article. For research articles, ACP prefers titles that highlight the scientific results/findings or implications of the study. I am not sure if this is the case here. I like the idea of mentioning "self healing" in the title (as this is a commonly known concept), but ideas like 'stabilising the ozone layer' are not well known and would not tell the reader much (before reading the paper).

There are general guidelines for ACP papers:

`https://www.atmospheric-chemistry-and-physics.net/policies/guidelines_for_authors.html`

I think the abstract is currently about 270 words which is longer than suggested by ACP. Consider shortening the abstract.

**Some minor issues**

- l. 21: explain what is "dangerous".

- l. 22: here you could mention transport of ozone.

- Fig. 1: What is the reason for the 'blue' areas in Fig. 1. b towards the polar regions?

- l 32.: citation for "generally"

- l. 33. the processes with an impact on $O_3$ should be discussed.

- l. 52: absorption of what?

- l. 104: I am not sure, but are you arguing about the $HO_x$ production from O here? This discussion could be more explicit.

- l. 107: "might contribute" – how can we (e.g. looking at model results) determine whether there is a contribution or not?

- l. 113: "some of the O" – this is a major point (see the discussion on $O_x$ above). Can you quantify "some"? Below (page 15) you say that "some" is practically all. That implies that R3 is not really a sink of $O_3$. More discussion?

- l. 115: "leakage of O" – I agree. But this is dominated by the catalytic cycles that are ignored in the Chapman theory.

- l. 116: stratospheric versus tropospheric ozone: "grows with altitude towards the surface" is unclear.

- l 125: can you better quantify "some amount"

- l. 127: you emphasise the *magnitude* here – but doesn't this involve the magnitude of the catalytic $O_x$ loss terms?

- l. 128: regarding the "magnitude", it would be good to have a good estimate of the chemical loss rates appropriate here.

- l. 131: transport would be helpful here.

- l. 131: here and elsewhere: does "column ozone" mean total column ozone' or the 'partial column ozone above the altitude in question'

- l. 138: this is true only if transport of ozone can be neglected.

- l 142: citations for "previous studies"

- l. 148: (Eq. 1) how are the $P$ and $L$ terms that are used here calculated?

- l. 150: how is this "basic state" calculated/determined?

- l. 160: Fig. 1 shows the polar latitudes nonetheless.

- l. 161: are there recent citations?

- l. 171: the "deep tropics" are mentioned here: is this the suggested range of validity of the present analysis? Is there any other reason for discussing a particular atmospheric region?

- l. 176: I do not agree that the "overall shape of the ozone layer" is described by the Chapman theory (see Fig. 5c in this paper and Fig. 2 in this review).

- Fig. 3: panels a) and b) look similar but I am not convinced that they are. I suggest a difference plot. And what about summer/winter conditions? They should be considered as well. And there is a substantial increase in reaction rates with altitude.

- l. 191: citations for "is attributed"?

- l. 195: I like the idea behind Eq. (2), but I am afraid it is not clear how the "fully interactive" used here is calculated. Below it looks like a calculation based on the Cariolle scheme, but this should be clear when the term is introduced.

- l. 204: convert from mixing ratios to number density – but why? What is the reason for making this step?

- l. 207: "quasi-steady state" – is this assumption justified?

- l. 217: does this mean that you investigate only tropical ozone here?

- l. "uniform reduction" is not what is observed in the real atmosphere regarding tropical ozone depletion (WMO, 2022).

- l. 229: How is $G_0$ defined? How is this quantity derived? Also what altitude is $z_1$? Why has a value of 60 km been chosen?

- l 251: $G_0$ could be defined as a perturbation in mixing ratio – would this not be better? At least it should be considered, what a constant value of $G_0 = 10^8$ molec cm$^{-3}$ in ozone density means for mixing ratio.

- l. 268: why only half? Photochemical adjustment gives you the sign of the change, but why is it half?

- l. 273: "entire stratosphere" – also when transport is important?

- l. 89: is there a latitude range for the validity of the "40 km" statements?

- l. 297: I do not agree with this assessment if the Chapman cycle. See for example Fig. 5c in this manuscript. How does the comparison (Chapman vs. real world) look for ozone at other latitudes and seasons?

- l 333: argument for "photochemical equilibrium"?

- l. 334: the atmosphere is not isothermal – why is this assumption necessary? It only affects the chemical rate constants – is this correct?

- l. 343: You could use a compilation of photochemical parameters (Burkholder et al., 2019) instead of a textbook. No major impact for the questions treated in the manuscript.

- l. 355: photolysis of ozone (R3) is seen here (and elsewhere in the manuscript) as a sink of ozone. This is not the case, if the $O_x = O + O_3$ family is considered (independent of Chapman). This is explained below in terms of a null cycle for ozone (l. 356, l. 371), which reiterates many of the ideas of $O_x$; the loss of $O_x$ is through R4. It is not clear to me why the $O_x$ is not used, but then reintroduced (it seems) through the ozone null cycle.

- l. 385: try `\large(` for larger brackets

- l. 387: an equation can not be negative (only a term in an equation)

- l. 396: how is the situation for other months?

- Eqs. 14 and 15: are these equations valid at the pole?

- l. 426: is "above 40 km" valid for all altitudes?

- l. 429: why is it not possible to calculate $P$ and $L$ in a chemical model?

- l. 451: suggest separating transport and catalytic chemistry – these two processes can be rather different.

- l. 456: you say "tropics" here – is the analysis also valid outside the tropics.

- l. 458: photolysis of ozone mostly drives the null cycle; see above.

- l. 465: here (and elsewhere in the manuscript): what is meant by "column ozone"? I assume, partial column ozone above the level is question. And not total column ozone. I suggest being a bit more precise here.

- l. 482: "terminates" – coming from above of from below?

- l. 493: "approximating". Is this true for all conditions discussed here? E.g. at all seasons? I am also not sure that the approximation is needed.

- l. 554/555: this text is similar to the text in lns. 498/499.

- l. 581: "all the way to the surface" – isn't this extrapolation a bit dangerous given the fact that tropospheric chemistry is neglected?

- l. 586: Do you have a citation for "surprising"?

- l. 595: I am confused here. What is the major difference between the chemistry of MOBIDIC and the Cariolle scheme? Is there a chemical difference? Or is it mainly transport?

- l. 639: incomplete citation

- l. 614: You say "tropics" here: is the paper on tropical ozone?

- l. 619: "down to the surface": this sounds very speculative; the analysis presented here is not for tropospheric altitudes in many respects.

- l. 620: The MOBIDIC code and the employed coefficients of the Cariolle scheme (Eq. 1) should also be available.

- l 645: which journal?

- l. 705: citation should contain pages or an electronic id.

- l. 709: is this citation correct? Do you mean WMO (2018)?

**References**

Burkholder, J. B., Sander, S. P., Abbatt, J. P. D., Barker, J. R., Cappa, C., Crounse, J. D., Dibble, T. S., Huie, R. E., Kolb, C. E., Kurylo, M. J., Orkin, V. L., Percical, C. J., Wilmouth, D. M., and Wine, P. H.: Chemical kinetics and photochemical data for use in atmospheric studies, Evaluation Number 19, JPL Publication 19-5, URL `http://jpldataeval.jpl.nasa.gov`, 2019.

Cariolle, D. and Teyssèdre, H.: A revised linear ozone photochemistry parameterization for use in transport and general circulation models: multi-annual simulations, Atmospheric Chemistry and Physics, pp. 2184–2196, 2007.

Fleming, E. L., Jackman, C. H., Stolarski, R. S., and Douglass, A. R.: A model study of the impact of source gas changes on the stratosphere for 1850–2100, Atmos. Chem. Phys., 11, 8515–8541, https://doi.org/10.5194/acp-11-8515-2011, 2011.

Fleming, E. L., George, C., Heard, D. E., Jackman, C. H., Kurylo, M. J., Mellouki, W., Orkin, V. L., Swartz, W. H., Wallington, T. J., Wine, P. H., and Burkholder, J. B.: The impact of current $CH_4$ and $N_2O$ atmospheric loss process uncertainties on calculated ozone abundances and trends, J. Geophys. Res., 120, 5267–5293, https://doi.org/10.1002/2014JD022067, 2015.

Portmann, R. W., Daniel, J. S., and Ravishankara, A. R.: Stratospheric ozone depletion due to nitrous oxide: influences of other gases, Phil. Trans. R. Soc. B, 367, 1256–1264, https://doi.org/10.1098/rstb.2011.0377, 2012.

WMO: Scientific assessment of ozone depletion: 1989, Report No. 20, Geneva, Switzerland, 1990.

WMO: Scientific assessment of ozone depletion: 2018, Global Ozone Research and Monitoring Project–Report No. 58, Geneva, Switzerland, 2018.

WMO: Scientific assessment of ozone depletion: 2022, GAW Report No. 278, Geneva, Switzerland, 2022.

---

## Author Comment (AC1)

**Response to Reviewers: "Beyond self-healing: Stabilizing and destabilizing photochemical adjustment of the ozone layer" (EGUSPHERE-2024-147)**

Aaron Match, Edwin P. Gerber, and Stephan Fueglistaler

June 10, 2024

We thank the reviewers for their thoughtful investigations of the first submitted version of the manuscript, and for their encouragement and critical comments, which have encouraged us to make several major revisions to the manuscript. These revisions address valid concerns raised by the reviewers while preserving and strengthening our main results. Throughout this Response to Reviewers, reviewer comments will be in black and our author comments will be in blue.

Before proceeding to line-item responses, we summarize the major revisions we have made to the manuscript. The most significant change is in the formulation of the model used to construct our simple theory of photochemical sensitivity. The previously-submitted draft showed that the Cariolle v2.9 linear ozone model predicted both photochemically stabilizing and destabilizing regimes. The previously unexplained destabilizing regime was reproduced in the Chapman Cycle, which was then subject to theoretical analysis to derive constraints on the transition from destabilization to stabilization. Although the Chapman Cycle was successful in reproducing the photochemical regimes, the suitability of the Chapman Cycle for representing the stratospheric ozone layer was validly questioned by Reviewer #2, based on the fact that the Chapman Cycle omits the leading-order sinks of ozone from catalytic cycles and transport. It is indeed puzzling why the Chapman Cycle should have done so well, a puzzle that we believe is resolved in the revised draft.

In the revised draft, we formulate an intermediate-complexity photochemical system, the *Chapman+2 model*, which augments the Chapman Cycle with generalized destruction of O and $O_3$ representing sinks of odd oxygen from catalytic cycles and (highly idealized) transport. The Chapman+2 model captures the leading-order sinks of odd oxygen in the tropical stratosphere, while retaining a level of simplicity that facilitates theoretical insight. We have re-cast the theoretical analysis of the revised paper (Section 4 onwards) using the Chapman+2 model.

Interestingly, the photochemical regimes of the Chapman+2 model are described by a similar theory to those in the Chapman Cycle itself, a result that is shown in the Discussion

to be a consequence of the fact that the Chapman Cycle and the catalytically O-damped limit of the upper stratosphere in the Chapman+2 model both have a loss of odd oxygen that is rate-limited in the availability of atomic oxygen, and is therefore proportional to the photolysis rate of $O_3$ (Section 7.1 of the revised manuscript).

The Chapman+2 model clarifies a conceptual point that was raised by Reviewer #2: we refer in the original version of the manuscript to a photolytic sink of $O_3$, even though the photolysis of $O_3$ is well known to preserve odd oxygen. Yet, both the Chapman Cycle limit and the O-damped limit have a photolytic sink of $O_3$. This can be understood because, in these limits, the loss of odd oxygen is limited by the availability of atomic oxygen that is primarily produced by photolysis of $O_3$. Physically, the photolytic sink of $O_3$ in the O-damped limit results because catalytic sinks of odd oxygen are rate-limited in atomic oxygen, and therefore speed up when enhanced photolysis of $O_3$ repartitions odd oxygen in favor of atomic oxygen. We note this in the revised Introduction as follows:

"*This standard explanation neglects the sensitivity of the ozone sink to perturbations in UV fluxes. One reason that the ozone sink has been neglected is that photolysis of $O_3$ is typically considered to have a neutral effect on the ozone layer. This is because photolysis of $O_3$ liberates an atomic oxygen that typically bonds with $O_2$ to reform $O_3$, completing a null cycle. Because the atomic oxygen is prone to reforming $O_3$, it is standard to analyze ozone photochemistry in terms of odd oxygen ($O_x \equiv O + O_3$), which is longer-lived than ozone and preserved by photolysis of $O_3$ (e.g., Jacob, 1999; Brasseur and Solomon, 2005). However, this null cycle does not close perfectly, and leakage from the null cycle occurs if the atomic oxygen bonds with $O_3$ (as in Chapman, 1930), or, as happens more often, with catalysts of $O_3$ depletion such as $HO_2$ or $NO_2$ (Bates and Nicolet, 1950; Crutzen, 1970). Although small compared to the null cycle, this leakage means that UV can drive a photolytic sink of $O_3$. In other words, although odd oxygen is preserved by photolysis of $O_3$, the sink of odd oxygen is nonetheless sensitive to such photolysis.*"

By reformulating our analysis in terms of the Chapman+2 model from the Chapman Cycle, we have addressed Reviewer #2's concerns about neglecting transport, about using the Chapman Cycle outside of its range of applicability, and about considering our analysis in light of the $O_x$ family. More detail will be offered below about the specific ways in which this new formulation addresses these valid questions.

Note that the Chapman+2 model is also being used in a closely-related preprint submitted to ACP (Match et al., 2024), which explains the signature interior maximum of ozone number density. That preprint explains the formulation of the Chapman+2 model in more detail, and characterizes its behavior into different regimes by the limiting behavior of the sink. The present revised manuscript nonetheless provides a stand-alone introduction to the Chapman+2 model, and all key results from the Chapman+2 model are described completely within.

Additional major changes to the manuscript include that we now report photochemical destabilization in an additional linear ozone model, LINOZ, based on the University of California Irvine chemical transport model (UCI CTM) driven by transport fields from NASA

GISS ModelE. The photochemical sensitivity of LINOZ quantitatively matches that of the Cariolle v2.9 linear ozone model, adding robustness to our key new finding of photochemical destabilization above 40 km in the tropical stratosphere. We also now consider photolysis of $O_3$ in the Chappuis bands, extending our numerical solution up to 800 nm from a previous largest wavelength of 320 nm, which does not qualitatively change our results. In terms of presentation, background information on self-healing has been streamlined into the Introduction.

**1 Reviewer #1**

**General Comments:**

This paper examines the concept of "self-healing" and "reverse self-healing" and proposes a more general mechanism called "photochemical adjustment". This adjustment can be stabilizing or destabilizing. Stabilizing adjustment is simply showing that the original ozone perturbation is damped top-down. Destabilizing adjustment is more complex and depends on how the top-down enhancement of the UV flux interacts with the odd-oxygen loss process. The authors discuss how the magnitude of the ozone slant column determines if a photochemical adjustment is stabilizing or destabilizing. I also found this paper interesting in that the concept of "self-healing" has been around since Johnston (1972), but a quantitative theory on the "self-healing" process has not been examined in detail till this work. I found this paper to be clearly written. The concepts are laid out in a readable manner (if anything a bit redundant). This work will not only be useful for perturbation studies of the present atmosphere but also for early Earth chemistry studies.

I highly recommend this paper to be published!

We thank Reviewer #1 for their supportive review, and we agree that the time is ripe for a quantitative analysis of self-healing. Some of the redundancies in the previous draft have been streamlined.

**Specific Comments:**

**Section 1**   (no action needed)

The authors do a nice job of summarizing the discussion of "self-healing" in the literature.

**Section 2**   (no action needed)

The authors clearly describe the regimes where "photochemical adjustment" operates either in a stabilizing or destabilizing manner. Destabilization is not an expected result and can only be "distinguished by the magnitude of the response without a quantitative theory".

**Section 3**   (suggestion made)

This section uses a chemistry-climate model to examine the net ozone production rate (production-loss terms). E.g., if the net ozone production is positive for an overhead positive perturbation than the photochemical adjustment is destabilizing. Using the chemistry-climate model is the step needed to quantify the photochemical adjustment theory. It was nice to see you describe what the linear ozone model includes in Equation 1 and show the A6 equation plotted in Figure 3. It was also interesting that a simpler Chapman-only chemistry representation is consistent above 40km.

Interpretation of Figure 3. When one looks closely at the zero contour in Figure 3a and 3b, the altitude where destabilization starts is much lower in Figure 3a than in 3b suggesting the additional odd-oxygen families (NOx, HOx, ClOx, etc?) are important for defining this threshold. I don't believe you make this point in this section. You may also want to add a new Figure (3c) that shows a profile of equation A6 for both chemical mechanisms.

We have reformulated Figure 3 to include an additional linear ozone model, LINOZ, which agrees with Cariolle v2.9, and a reformulation of our theoretical model (described in the Introduction to this Response to Reviewers)—the Chapman+2 model. Even more than in the Chapman Cycle, the Chapman+2 model has a modestly higher altitude for its transition from destabilization aloft to stabilization below than the linear ozone models. The potential causes of this difference are manifold, and need not be restricted to catalytic chemistry. We discuss them in the following paragraph of the revised paper (Section 4.3 of the revised paper):

"The solar zenith angle has been approximated by the latitude (neglecting the diurnal cycle) to allow a global column-by-column comparison with the photochemical sensitivity of the linear ozone models during the equinoctial month of March (Figs. 3a,b). The Chapman+2 model reproduces the gross features of the linear ozone models, most importantly the destabilizing regime above 40 km in the tropics and the stabilizing regime below. The Chapman+2 model reproduces the latitudinal dependence in which the transition between the destabilizing regime and the stabilizing regime shifts up to higher altitudes at higher latitudes. As a caveat, the representation of transport-based damping of $O_3$ in the Chapman+2 model is not designed to represent the extratropics, for which transport is generally a source of ozone. However, transport only enters indirectly into the photochemical sensitivity of the Chapman+2 model through its effects on the basic state ozone, with $\kappa_{O_3}$ not appearing explicitly in Eqs. 15 and 16, so these results are not expected to be overly sensitive to the unrealistic aspects of transport."

**Section 4**   (suggestion made)

This is a very clever approach in equation (2) to derive "photochemical adjustment". One suggestion that would make reading your logic on photochemical adjustment easier would be to not include the equations in the sentences of the document (page 10) but delineate them as you did equation (3). This is purely a style suggestion.

We see the appeal of this stylistic suggestion, although we have opted to keep the relevant equations in-line in order to distinguish them by their algorithmic nature from the more fundamental indented numbered equations of the manuscript.

**Section 5**   (no action needed)

Very clearly written. Excellent job of taking the reader through each step of deriving the photochemical adjustment in the Chapman Cycle.

Thank you. As the reviewer will note, a very similar derivation is now presented for the Chapman+2 model, which we believe retains the clarity of the first version.

The main difference between Figure 4 and Figure 6 seems to be in the perturbation response to cooling (not ozone response to depletion). As you state in lines 450-453 "reverse self-healing is stronger in the Off-line Cariolle v2.9 emulator than in the Chapman Cycle, suggesting a role for non-Chapman stabilizing processes including catalytic chemistry and transport." It is nice to know that detail chemistry plays a role here!

And indeed the detailed chemistry omitted in our first version does seem to matter for reverse self-healing, as our new version of Figure 6 with the Chapman+2 model shows a realistically deep region of reverse self-healing.

**Section 8**   (suggestion made)

It is not clear to me how one can derive the role of transport in photochemical adjustment. Your comments in this section do not seem to add any clarity on how this could be derived. Can you expand on what "further work" would look like or can you at all estimate what the maximum role of transport would play in your current analysis?

The revised manuscript addresses the role of transport head-on by considering the Chapman+2 model, in which transport is represented in an idealized way as a damping of $O_3$, motivated by representing the tropical lower stratosphere in which transport upwells ozone-poor air from below. This idealized treatment of transport is now part of the basic state of our model, and is responsible for the dominant sink of $O_3$ in the tropical lower stratosphere below the ozone maximum. Our more open-ended discussion of the role of transport in the Discussion now focuses on a more refined question, which will hopefully be clearer to the reader (Section 7.1 of the revised manuscript):

"*It has been suggested that self-healing and reverse self-healing are damped by transport, i.e., that photochemistry would stabilize a larger fraction of the UV anomaly if not for the damping effects of transport (Solomon et al., 1985). The fraction of photochemical stabilization is important because, in a hypothetical world with perfect photochemical stabilization, ozone-depleting substances that destroy ozone aloft would add exactly the same amount of ozone back through self-healing, eliminating anthropogenic impacts on the ozone layer. Thus, deviations below perfect stabilization lead to risks from ozone-depleting substances and benefits from their mitigation.*

*Does transport suppress the fraction of photochemical stabilization, as has been argued,*

*by damping $O_3$ anomalies in the lower stratosphere? The answer is not clear-cut when the basic-state effects of the transport are accounted for, because although $O_3$ damping by transport certainly damps $O_3$, it also creates the $O_3$-damped regime of the lower stratosphere that is prone to photochemical stabilization. Transport introduces the possibility that $O_3$ depletion aloft can allow more ultraviolet fluxes below that speed up photochemical equilibration in the lower stratosphere, shifting down the transition altitude between the $O_3$-damped limit and the O-damped limit. Reflecting this nuance, increasing $\kappa_{O_3}$ in the Chapman+2 model actually increases the fractional stabilization of the UV-locked $O_3$ anomaly from stratospheric cooling, while reducing the fractional stabilization of the UV-locked $O_3$ anomaly from $O_3$ depletion. Further work could clarify whether transport amplifies or suppresses this fraction of photochemical stabilization.*"

**Section 9** Nice summary. It would be very interesting to complete this analysis in an early Earth atmosphere.

Thank you, we agree that such an analysis would be fascinating.

**2    Reviewer #2**

We thank Reviewer #2 for their clear and constructive review, which has led to major improvements in the manuscript. We first respond broadly to the major points raised, and then provide detailed in-line responses throughout.

I like the general idea of the paper, namely a focus on the issue of "self-healing" and to look into the detailed chemical mechanisms involved. And these mechanisms are indeed discussed in detail. But I also have reservations about the study. First, I think that the paper is not well formulated in many respects; I suggest a "methods" section where the employed models are better described and introduced. And the available literature at this point is rather old.

We are pleased to hear that the reviewer broadly likes the general idea of the manuscript. We agree that one main issue of the methods must be clarified, which is the status of the chemistry-climate models and linear ozone models in our argument. Our argument uses the coefficients of linear ozone models (Cariolle v2.9 and now also LINOZ), but our analysis does not involve running the "parent" models directly (the parent models being MOBIDIC and now also UCI CTM/NASA GISS ModelE). We believe that clarification of this point will go a long way towards clarifying our methods, which we have opted stylistically to still introduce as they are used rather than up front all at once. Regarding linear ozone models, we now say (Section 2), "*This exact battery of tests has been performed in chemical transport models by previous studies that developed linear ozone models, two independent sets of coefficients from which are used throughout this study: Cariolle v2.9 (Cariolle and Déqué, 1986; Cariolle and Teyssèdre, 2007) and LINOZ (McLinden et al., 2000).*"

Second, the paper sounds almost like a global study (e.g. abstract), but throughout

the paper the tropics are mentioned. (But not the mod- latitudes). So is this a paper on tropical ozone? The neglect of transport seems to limit the range of applicability.

We thank the reviewer for recommending that we clarify the geographic scope. Our results have different geographic scope depending on the tool used (stratosphere/troposphere, tropics/extratropics). The linear ozone model results are global, because they are run from a model with full chemistry, although inheriting any idiosyncrasies from their parent models. To emphasize this global scope, we now say (Section 2 of the revised manuscript), "*We have globally analyzed the $A_6$ coefficients in two models.* "

These global results are interpreted with the use of a simplified photochemical model, which in the revised manuscript is the Chapman+2 model. The Chapman+2 model is restricted in geographic scope in two ways: its chemistry is designed to represent that in the stratosphere and not in the troposphere, and its idealized representation of transport is specifically designed to represent the tropical lower stratosphere. When introducing the Chapman+2 model, we constrain its scope by noting (Section 4.1 of the revised manuscript): "*The chemical reactions that comprise the damping in the Chapman+2 model are intended to provide an idealized representation of the global stratosphere, while excluding both heterogeneous chemistry in the polar stratosphere and tropospheric chemistry. The simplified representation of transport is intended to provide an idealized representation of only the tropical lower stratosphere. These geographical caveats will be considered when using the Chapman+2 model to interpret the global results from the linear ozone models.*"

We return to these caveats when comparing a global result from the Chapman+2 model with the linear ozone models (Section 4.3 of the revised manuscript):

"*Explicit calculations of the photochemical sensitivity of the Chapman+2 model are shown in Fig. 3c. The solar zenith angle has been approximated by the latitude (neglecting the diurnal cycle) to allow a global column-by-column comparison with the photochemical sensitivity of the linear ozone models during the equinoctial month of March (Figs. 3a,b). The Chapman+2 model reproduces the gross features of the linear ozone models, most importantly the destabilizing regime above 40 km in the tropics and the stabilizing regime below. The Chapman+2 model reproduces the latitudinal dependence in which the transition between the destabilizing regime and the stabilizing regime shifts up to higher altitudes at higher latitudes. As a caveat, the representation of transport-based damping of $O_3$ in the Chapman+2 model is not designed to represent the extratropics, for which transport is generally a source of ozone. However, transport only enters indirectly into the photochemical sensitivity of the Chapman+2 model through its effects on the basic state ozone, with $\kappa_{O_3}$ not appearing explicitly in Eqs. 15 and 16, so these results are not expected to be overly sensitive to the unrealistic aspects of transport.*"

We also now provide a recommended interpretation of the tropospheric component of our vertical profiles using the Chapman+2 model, which should be interpreted as illustrating the extended properties of the stratospheric solution rather than the tropospheric response (Section 4.4 of the revised manuscript): "*The Chapman+2 model is designed to represent photochemistry in the stratosphere but not the troposphere, so while full pro-*

*files are shown down to the surface in order to provide a complete characterization of the stratospheric regime, these should not be considered to represent the true response in the troposphere."*

Third, I do not agree that the Chapman cycle constitutes a good approximation of the state of the ozone layer; the loss term of ozone in the atmosphere is at no altitude dominated by the Chapman reaction R4 (Fig. 2).

We agree (see also our opening remarks in this response) that it is not obvious why the Chapman Cycle should constitute a good approximation of the sensitivity of the ozone layer to perturbations (and that the original manuscript was lacking in this respect). That the Chapman Cycle reproduced the sensitivity of the ozone layer so well is surprising because the Chapman Cycle is known to omit the leading-order sinks of odd oxygen from catalytic cycles and transport, and the reviewer has expressed legitimate skepticism at this result. In the revised version, we have reformulated our analysis in terms of a Chapman+2 model that includes an idealized representation of sinks of O and $O_3$ to representation catalytic cycles and tropical lower stratospheric transport. This reformulation incorporates these leading-order sinks while retaining a level of simplicity that facilitates theoretical insight, and we now can explain why the Chapman Cycle performs so well at reproducing the full response from comprehensive chemistry models; it is because the O-damped limit of the upper stratosphere has a photolytic sink of $O_3$ (i.e., a sink that is rate-limited in atomic oxygen), and the Chapman Cycle does, too. This result is now covered in the Discussion (Section 7.1 of the revised manuscript).

Finally, the assumptions on stratospheric cooling and ozone depletion are likely too crude (no time period given, no altitude dependence assumed) so that the question arises what can be learned about the real atmosphere based on such assumptions.

It is well known that heterogenous chemistry and stratospheric cooling impose perturbations with nontrivial vertical structure on the ozone layer. Our goal is not to refine detailed projections of the ozone layer response to these perturbations, which are simulated well in models that already include the physics described in our paper. Rather, we seek to provide new physical understanding of this "backbone" of ozone photochemistry against which CFCs or stratospheric cooling operate. We now state this aim more clearly (Section 3 of the revised manuscript): *"We assess the response of tropical ozone to highly-idealized forcing by ozone depletion (e.g., from ozone depleting substances) and stratospheric cooling (e.g., from the direct radiative effects of $CO_2$). The response to such perturbations is generally known to have nontrivial vertical structure (e.g., Fig. 1), although this response already includes the effects of photochemical adjustment, whose structure we seek to isolate. In order to illustrate how photochemical adjustment can induce nontrivial vertical structure even in response to uniform perturbations, we consider constant (or piecewise-constant) perturbations.*

We subsequently further caveat the response in the troposphere (Section 3 of the revised manuscript): *"This cooling is imposed uniformly, even extending into the troposphere, where the response is small but should not be considered as a realistic response to elevated*

*$CO_2$, which actually warms the troposphere.*"

I summary, I think the paper needs work and improvement. The assumptions (and range of applicability) should be clear and there should be a clear message.

We thank the reviewer again for their careful and constructive criticism, which we believe has been addressed through these major revisions, leading to an improved paper. We now address each comment in more detail.

**Comments**

**Neglecting stratospheric transport of ozone**   Clearly stratospheric ozone is strongly influenced by transport (Fig. 1 of this review); this is acknowledged by the authors of the paper. However, in the main body of the paper this is only noted in passing (and not at all in the introduction) and not really employed in the text. I have noted that there is section 8.1. But in my opinion, the study needs to be clear about in which regions (due to transport) the employed analysis is not applicable. At the moment the study reads like a "photochemistry only" study, where transport is inserted as an "add on".

As noted above, the most important direct effects of transport on odd oxygen in the tropical stratosphere, i.e., its damping of ozone in the tropical lower stratosphere, are now considered explicitly in the Chapman+2 model. This enriches our analysis, e.g., by making it clear that in a strictly transport-dominated regime, which approximates the situation in the tropical lower stratosphere, ozone is controlled only by its photolytic source and would not generally be expected to produce photochemical destabilization. Consistent with prior understanding, transport of odd oxygen is not significant above about 26 km, where both photochemical stabilization and destabilization remain possible and in fact are found to occur in the Chapman+2 model and in the linear ozone models.

**Chapman cycle**   The paper makes the point that the "Chapman model" explains the gross features of the ozone layer. I am sorry, but I cannot agree. The "Chapman model" is known to be incorrect since decades. At no altitude, the Chapman reaction R4 dominates ozone loss (Fig. 2 of this review; see also Portmann et al. (2012) and the textbooks cited in the manuscript). The total column ozone (referred to in this paper) globally (and seasonally resolved) would look very different than observed, if there was only the "Chapman model". The problem is already visible in Fig. 5C (of the paper), but how would look a comparison of ozone (Chapman vs. observations) in mid-latitudes in winter? Or in the polar regions.

The authors have a good point in emphasising the wave length resolved variation of penetration of radiation throughout the stratosphere. I understand the idea of the study is to quantify the response of the (tropical?) ozone layer to ozone depletion aloft, but in my understanding this requires to get the ozone loss terms (and their dependency on photolysis) right—the "Chapman model" cannot do this.

Finally, I cannot see a conceptional advantage of using the "Chapman mode" —the catalytic cycles that impact ozone loss essentially speed up reaction R4 (compared to a

[Figure]

Figure 1: Chemical and dynamical control of ozone in the stratosphere. (from WMO 1990).

[Figure]

Figure 2: Figure 2: The left panel shows the relative global mean ozone loss rates by chemical family computed for 2000 levels of source gases by the NOCAR two- dimensional model. The right panel shows the global mean ozone profile, which highlights the ozone layer maximum in the middle stratosphere. (from Portmann et al., 2012).

pure "Chapman model") so why is it not possible to go this route?

As noted above, the reviewer has legitimately pointed out limitations of the Chapman Cycle, and recommended that we consider augmenting its ozone loss to make it more realistic. This is exactly what we have done through our major revisions, whereby we have incorporated sinks of ozone that are not included in the Chapman Cycle, with our analysis now performed in the Chapman+2 model. Interestingly, our analysis of the Chapman+2 model has shown that qualitatively different limits for ozone photochemistry result depending on whether the augmented loss destroys O, $O_3$, or $O+O_3$ (i.e., speeds up R4). Each of these limits is of theoretical and pragmatic interest, and the manuscript has been enriched by considering them, while the fundamental results remain the same but with stronger justification thanks to the more realistic model.

**Range of validity of the results** The paper should be clear about the range of validity of the results presented here. If I read the abstract (and the title) it looks like a global analysis. But it is not. Clearly, heterogeneous chemistry (and thus polar ozone loss) is neglected (which would be available in the linear scheme Cariolle and Teysse'dre, 2007), this is okay. But nonetheless, Fig. 1 extends to the poles.

Clearly tropospheric ozone chemistry is not the point here, this paper is on stratospheric processes. And certainly there is no 10 K cooling in the troposphere. In spite of this, Figs. 1, 6 and 7 extend to the ground.

Moreover, I am not convinced that the processes discussed here are valid for the entire stratosphere. Sometimes, the tropics are mentioned (e.g., Fig. 5c is only for the tropics) and particular effects are only discussed for the tropics (e.g., line 456). Does this mean that the analysis presented here is only for the tropics? If yes it should be stated. And how are the tropics defined here? If both the tropics and the extra-tropics are discussed here, there are likely different processes relevant for these altitudes—do you disagree?

Finally, in the tropics, the mean reaction rate between $\approx$ 20-40 km increases by more than an order of magnitude —I think this process is relevant for the arguments on chemical processes put forward here and should not be ignored.

As noted above, in the revised manuscript, we have clarified the range of validity of the results, which depends on the tools used in each section. The linear ozone model results are global. The Chapman+2 model results have a photochemistry scheme that is confined to the stratosphere (not the troposphere), and an idealized representation of transport that is confined to the tropics (not the extratropics).

**The employed assumptions on ozone loss and stratospheric cooling** This study uses assumptions on ozone loss and stratospheric cooling. These assumptions are supposed to be somewhat realistic to be useful and applicable to the real atmosphere. First, suggesting a change in ozone or temperature without giving a time period is not very useful. If you are looking for numbers of changes that are realistic for particular time periods, WMO

(2022) might be helpful. The strongest ozone change is at the poles (not treated here) and in the upper stratosphere.

Second, assuming a uniform (altitude independent) change of ozone throughout the atmosphere is not realistic (WMO, 2022) and this assumption is even worse for temperature (sorry, but you seem to assume 10 degree reduction of temperature at ground level . . . ).

For convenience, we repeat how we addressed this comment, as was noted in earlier in this Response to Reviewers: It is well known that heterogenous chemistry and stratospheric cooling impose perturbations with nontrivial vertical structure on the ozone layer. Our goal is not to refine detailed projections of the ozone layer response to these perturbations, which are simulated well in models that already include the physics described in our paper. Rather, we seek to provide new physical understanding of this "backbone" of ozone photochemistry against which CFCs or stratospheric cooling operate. We now state this aim more clearly (Section 3 of the revised manuscript): "*We assess the response of tropical ozone to highly-idealized forcing by ozone depletion (e.g., from ozone depleting substances) and stratospheric cooling (e.g., from the direct radiative effects of $CO_2$). The response to such perturbations is generally known to have nontrivial vertical structure (e.g., Fig. 1), although this response already includes the effects of photochemical adjustment, whose structure we seek to isolate. In order to illustrate how photochemical adjustment can induce nontrivial vertical structure even in response to uniform perturbations, we consider constant (or piecewise-constant) perturbations.*"

We subsequently further caveat the response in the troposphere (Section 3 of the revised manuscript): "*This cooling is imposed uniformly, even extending into the troposphere, where the response is small but should not be considered as a realistic response to elevated $CO_2$, which actually warms the troposphere.*"

**Ozone (O3), O and the Ox family** The analysis in the submitted manuscript does not involve a discussion of the Ox = O3 + O family. The textbook knowledge (conventional wisdom) states that it does not matter whether O or O3 is lost through chemistry, but only if a member of the family Ox = O3 + O is lost. That is, reaction R4 (in the manuscript) is an Ox loss, but not R3. But in contrast, R3 is stated in the manuscript as a loss of ozone. The reason for using the concept of an Ox family is of course that reactions R2 and R3 are very fast (at least by more than an order of magnitude) compared to all other reactions of importance in the stratosphere. This point seems to be worked out again at the bottom of page 15 in the manuscript, i.e. the null cycle from R3 to R2. It is of course okay to disagree with the textbook knowledge on the Ox family but I do not recommend to ignore it. If you do something else here I suggest justifying it against the Ox family concept. As I read the paper, R3 is considered as a loss of ozone—I do not think this is true, with consequences for the arguments put forward in the manuscript. But I am happy to be convinced otherwise.

However, I think that all the other catalytic cycles impacting stratospheric ozone (and their dependence of the actinic flux at particular wavelengths) should not be ignored.

A linchpin of this manuscript is the idea that photolysis of $O_3$ can increase the sink of $O_3$. We appreciate the reviewer's encouragement to contextualize our argument compared to the textbook understanding. This idea remains consistent with the concept of odd oxygen, insofar as the sink of ozone from photolysis of $O_3$ is only a small leakage compared to a much larger null cycle between R2 and R3. Consistent with the idea of odd oxygen, most photolysis of $O_3$ does not destroy odd oxygen; however, what is often missed when thinking about odd oxygen is that in key parts of the stratosphere, most of the sink of odd oxygen requires photolysis of $O_3$ to produce atomic oxygen that is rate-limiting for the sink of odd oxygen. This is true for the Chapman Cycle and also true for the O-damped regime considered in the revised manuscript, in which the sink of odd oxygen is rate-limited in O.

The revised manuscript clarifies these points in several places:

In the Introduction of the revised manuscript: "*This standard explanation neglects the sensitivity of the ozone sink to perturbations in UV fluxes. One reason that the ozone sink has been neglected is that photolysis of $O_3$ is typically considered to have a neutral effect on the ozone layer. This is because photolysis of $O_3$ liberates an atomic oxygen that typically bonds with $O_2$ to reform $O_3$, completing a null cycle. Because the atomic oxygen is prone to reforming $O_3$, it is standard to analyze ozone photochemistry in terms of odd oxygen ($O_x \equiv O + O_3$), which is longer-lived than ozone and preserved by photolysis of $O_3$ (e.g., Jacob, 1999; Brasseur and Solomon, 2005). However, this null cycle does not close perfectly, and leakage from the null cycle occurs if the atomic oxygen bonds with $O_3$ (as in Chapman, 1930), or, as happens more often, with catalysts of $O_3$ depletion such as $HO_2$ or $NO_2$ (Bates and Nicolet, 1950; Crutzen, 1970). Although small compared to the null cycle, this leakage means that UV can drive a photolytic sink of $O_3$. In other words, although odd oxygen is preserved by photolysis of $O_3$, the sink of odd oxygen is nonetheless sensitive to such photolysis.'*

Later (Section 4.3 of the revised manuscript): "*The sensitivities of the photolytic source and sink are weighted by the leakage from the null cycle R2$\rightleftharpoons$R3, $\epsilon$, which measures the strength of the photolytic sink. When $\epsilon$ is zero, there is no photolytic sink of odd oxygen, leading to strict photochemical stabilization ($\frac{\partial(P-L)}{\partial \chi_{O_3}} < 0$). When $\epsilon > 0$, there is a photolytic sink of odd oxygen from some combination of the Chapman sink (captured in the term $k_4\overline{[O_3]}$) and the damping of atomic oxygen (captured in the term $\kappa_O/2$), and photochemical destabilization becomes possible, although not guaranteed. The leakage itself is small; for example, at 45 km, $\epsilon \sim O(10^{-2})$, meaning that out of 100 photolysis events of ozone, 99 will proceed into the null cycle and only 1 will leak into a sink of odd oxygen. However, photochemical destabilization nonetheless occurs at this altitude because $J_{O_3}$ at 45 km is six orders of magnitude more sensitive to perturbations in overhead column $O_3$ than is $J_{O_2}$.*"

**Model description and documentation** The paper uses the MOBIDIC model; I would not call this model a "chemistry- climate model" —it is a two-dimensional model;

("chemistry-climate model" sounds a bit like CMIP, which is misleading). As far as I understand the paper, the chemistry scheme used in MOBIDIC is the Cariolle scheme (Eq. 1) ? so I suggest to make this very clear in the paper. (Or the other way around, if I am not correct here). Further, the cited documentation on MOBIDIC is decades old (1985) and what is cited is not an extensive description of the model (but rather a conference contribution). (Is there a more recent description of the model?) It is also not clear from the paper, why for such two dimensional model calculations one needs to use the linear Cariolle scheme, rather than performing a full chemistry simulation. Regarding two-dimensional models, there were other options (e.g. Fleming et al., 2011, 2015, but of course there are other alternatives); I do not think that computational issues are an argument today (for such studies). Did the authors consider testing the performance of the Cariolle scheme against a full chemistry simulation?

I was indeed shown (Meraner et al., cited in the paper) that the annual mean total ozone column and the tropical ozone profile of MOBIDIC agree well for linear and the explicit chemistry schemes. However, other issues remain. For the pur- pose of the present study, I think that in particular the partial ozone column above some altitude is important. Further, is the performance outside of the tropics relevant here? If not, this needs to be explicitly discussed. If I understand correctly, MOBIDIC has issues with HNO3, especially in high latitudes in summer is this correct?

Also, MOBIDIC and the Cariolle scheme have certainly evolved over time during the decades since the cited reference in 1985. I would be good to have at least some information on the new parameters of the Cariolle scheme (see also below). Further, in the paper there is a discussion of a "fully interactive" calculation (see also below); there is some description of the calculation in the paper, but I suggest to make it very clear right from the start, which calculations and which models are used here (and which are not used, no tropospheric chemistry, no heterogeneous chemistry). Perhaps there could be a methods section in the paper, where such things are discussed?

We appreciate the reviewer noting that the manuscript generated some confusion about the usage of modeling tools. The revised manuscript clarifies that what we use in this study are the coefficients of a linear ozone model that is linearized with respect to MOBIDIC. The reviewer has rightfully pointed out that "chemistry-climate model" might evoke the wrong idea for what MOBIDIC is. MOBIDIC can also be aptly described as a chemical transport models, i.e., a chemistry model driven by prescribed transport, where the dynamical fields come from the model ARPEGE-Climat (Déqué et al., 1994). We have revised our language to refer to it as such.

The most recent description of the Cariolle v2.9 linear ozone model, which we use, is provided in Cariolle and Teyssèdre (2007), which we cite. Per the reviewer's recommendation, we have also tested the performance of the Cariolle v2.9 linear ozone model against a different linear ozone model: the LINOZ scheme that was calibrated by linearizing the UCI CTM when driven by dynamical fields from NASA GISS ModelE, first described in McLinden et al. (2000). The two linear ozone models strongly agree throughout the globe

regarding the photochemical sensitivity of the ozone layer, and in particular for our key result of large photochemical destabilization above 40 km in the tropical stratosphere. this lends confidence that our results are not hampered by idiosyncrasies from a particular photochemical model. These models are described with the latest available citations.

We have opted to clarify the description of each method upon its use in the manuscript, rather than introducing our methods all at once in a methods section. This is because our methods are inextricably related to the logic of our argument, so key decisions in formulating our methodology would be challenging to understand if they were introduced prior to being logically demanded by our physical argumentation.

**Abstract and title**  ACP suggests that titles should be concise and consistent with the content and purpose of the article. For research articles, ACP prefers titles that highlight the scientific results/findings or implications of the study. I am not sure if this is the case here. I like the idea of mentioning "self healing" in the title (as this is a commonly known concept), but ideas like "stabilising the ozone layer" are not well known and would not tell the reader much (before reading the paper).

There are general guidelines for ACP papers:

`https://www.atmospheric-chemistry-and-physics.net/policies/guidelines_for_`
`authors.html` I think the abstract is currently about 270 words which is longer than suggested by ACP. Consider shortening the abstract.

We thank the reviewer for their advice on the title. After consideration, we request stylistic accommodation of our existing title, which is the "neutral" ACP style and achieves two key goals parsimoniously: it evokes the idea of self-healing but makes clear that we will go "beyond" the state-of-the-art for this topic, and it introduces the concept of photochemical adjustment, the name for which is sufficiently evocative that we believe it will not be confusing, and whose inclusion in the title will allow us to headline this new concept.

The abstract has been revised to 250 words.

**Some minor issues**

- l. 21: explain what is "dangerous". In the course of revisions, this language referring to UV radiation as dangerous was removed.

- l. 22: here you could mention transport of ozone. Transport is now mentioned in our Introduction when introducing the Chapman+2 model (e.g., in the Introduction of the revised manuscript): "*The region of photochemical destabilization above 40 km is then reproduced in a highly simplified model of the Chapman Cycle augmented with two generalized sinks of O and $O_3$ to represent catalytic cycles and transport—the Chapman+2 model* "

- Fig. 1: What is the reason for the "blue" areas in Fig. 1. b towards the polar regions? This is an interesting question. These extratropical ozone reductions in response to stratospheric cooling are beyond the scope of our use for that figure, so we do not explain them in the paper. Further work beyond the scope of this paper would be required to disentangle the relative roles of (e.g.,) photochemical adjustment, temperature-dependent changes in heterogeneous chemistry, BDC changes, tropopause changes, etc.

- l 32.: citation for "generally" This sentence is supported by the citations of the previous sentence; we have added "op. cit.".

- l. 33. the processes with an impact on O3 should be discussed. Reaction rates are now discussed: "Fig. 1b provides a canonical example of reverse self-healing in response to stratospheric cooling from the direct radiative effects of elevated $CO_2$, the cooling from which changes collisional reaction rates to generally increase $O_3$, which then leads to reductions in the tropical lower stratosphere (op. cit.)."

- l. 52: absorption of what? This language has been revised out, but it referred to absorption of UV by $O_2$ and $O_3$

- l. 104: I am not sure, but are you arguing about the HOx production from O here? This discussion could be more explicit. Yes, this minor mechanism referred to HOx production, although has been revised out for brevity when streamlining the introduction.

- l. 107: "might contribute" — how can we (e.g. looking at model results) determine whether there is a contribution or not? Discussion of the possible contributions of this minor mechanism have been removed for brevity, retaining the broader theoretical analysis of photochemical adjustment that explains its mechanisms using the Chapman+2 model.

- l. 113: "some of the O" — this is a major point (see the discussion on Ox above). Can you quantify "some"? Below (page 15) you say that "some" is practically all. That implies that R3 is not really a sink of O3. More discussion? We have clarified the leakage where it is discussed quantitatively: "*The leakage itself is small; for example, at 45 km, $\epsilon \sim O(10^{-2})$, meaning that out of 100 photolysis events of ozone, 99 will proceed into the null cycle and only 1 will leak into a sink of odd oxygen. However, photochemical destabilization nonetheless occurs at this altitude because $J_{O_3}$ at 45 km is six orders of magnitude more sensitive to perturbations in overhead column $O_3$ than is $J_{O_2}$.*"

- l. 115: "leakage of O" — I agree. But this is dominated by the catalytic cycles that are ignored in the Chapman theory. We agree that catalytic cycles are important,

- and are now explicitly included in the Chapman+2 model, making clear that leakage of O is dominated by these cycles.

- l. 116: stratospheric versus tropospheric ozone: "grows with altitude towards the surface" is unclear. Confusing language removed.

- l 125: can you better quantify "some amount" The leakage is quantified as noted above, although we note that when considering the qualitative possibility of photochemical destabilization, any amount of leakage opens up the possibility for photochemical destabilization, e.g., "*The photolytic sink of $O_3$ opens unconventional possibilities for the $O_3$ response to perturbations in overhead column $O_3$.*"

- l. 127: you emphasise the magnitude here—but doesn't this involve the magnitude of the catalytic Ox loss terms? Good question. Photochemical destabilization does not advertise itself through the sign of the response, so only manifests in magnitude changes. These magnitude changes are diagnosed through UV-locking experiments, as we do using the linear ozone models and the idealized model.

- l. 128: regarding the "magnitude", it would be good to have a good estimate of the chemical loss rates appropriate here. The magnitude is diagnosed using the linear ozone model and the Chapman+2 model, where the responses can be compared to the initial perturbations.

- l. 131: transport would be helpful here. We now consider the effects of idealized transport explicitly in the Chapman+2 model.

- l. 131: here and elsewhere: does "column ozone" mean total column ozone? or the "partial column ozone above the altitude in question" Good clarification point; in most such instances, we typically mean "overhead column ozone", and have updated our language accordingly.

- l. 138: this is true only if transport of ozone can be neglected. This is a good and subtle point. Intuitively, it would seem that a weakening of both photochemical P and L that led to an increase in P-L could nonetheless lead to a reduction in ozone by switching the regime to transport-dominated. For our purposes, the calibration of linear ozone models assumes that tendencies from transport are fixed under the perturbation (i.e., A1 is fixed), in which case the net change in P-L dictates the sign of the change in ozone concentration. We equip the curious reader to appreciate this subtle point by noting, "*To assess whether the ozone layer is photochemically stabilizing or destabilizing at a given location, we evaluate the sensitivity of the local net production rate of ozone (production minus loss) to a UV perturbation induced by a change in overhead column $O_3$, holding all else fixed (including tendencies from transport).*"

- l 142: citations for "previous studies" Citations added.

- l. 148: (Eq. 1) how are the P and L terms that are used here calculated? P-L are photochemical terms diagnosed from the chemical transport models and stored in the table of coefficients for the linear ozone models.

- l. 150: how is this "basic state" calculated/determined? The basic state is the climatological state of ozone in the chemical transport model when driven by the prescribed dynamical fields, with the source of those dynamical fields now noted explicitly: "*The first linear ozone model was described in Cariolle and Déqué (1986), from which the Cariolle v2.9 linear ozone model descended, with a major update to specify the 2D circulation from the ARPEGE-Climat model (Déqué et al., 1994) and to include heterogeneous chemistry in Cariolle and Teyssèdre (2007).*"

- l. 160: Fig. 1 shows the polar latitudes nonetheless. We have removed the disclaimer about heterogeneous chemistry, as it is relevant to part of the work, but not all of the work. We now specify that the linear ozone model analysis is global.

- l. 161: are there recent citations? We are not aware of recent citations describing MOBIDIC, although its results are now shown in our work to be robust when compared to the independent UCI CTM driven by NASA GISS ModelE fields

- l. 171: the "deep tropics" are mentioned here: is this the suggested range of validity of the present analysis? Is there any other reason for discussing a particular atmospheric region? We have clarified the varied geographical scopes of our tools, with the deep tropics subject to more intensive study because it is well-described by the Chapman+2 model used later.

- l. 176: I do not agree that the "overall shape of the ozone layer" is described by the Chapman theory (see Fig. 5c in this paper and Fig. 2 in this review). We now use the Chapman+2 model and have a separate paper submitted to ACP that considers the shape of the ozone layer in the Chapman Cycle and the Chapman+2 model.

- Fig. 3: panels a) and b) look similar but I am not convinced that they are. I suggest a difference plot. And what about summer/winter conditions? They should be considered as well. And there is a substantial increase in reaction rates with altitude. The essential message of Figure 3 is the sign of the photochemical sensitivity, i.e., the surprising photochemical destabilization above 40 km. This can be discerned by eye from the present plots. Other seasons are not shown, but support this message (superposed upon which is a seasonal cycle over the poles.) Subsequent quantitative analysis of photochemical adjustment supports more refined quantitative results.

- l. 191: citations for "is attributed"? This sentence was removed for brevity, obviating the need to repeat the standard set of citations for self-healing.

- l. 195: I like the idea behind Eq. (2), but I am afraid it is not clear how the "fully interactive" used here is calculated. Below it looks like a calculation based on the Cariolle scheme, but this should be clear when the term is introduced. We have clarified "Fully interactive" as follows: "*Photochemical adjustment is therefore quantified as the difference between a fully interactive simulation of ozone photochemistry, in which ozone and UV are coupled in the column through the photolysis rates, compared to one where the UV fluxes (and hence photolysis rates) are locked at their unperturbed values:*". Note that the concept of a fully interactive simulation transcends the details of how it is calculated in this section, as it is also calculated for the Chapman+2 model and could be calculated in any chemistry-climate model. This illustrates the broad applicability of the concept of photochemical adjustment.

- l. 204: convert from mixing ratios to number density— but why? What is the reason for making this step? As now noted, "*for convenience*". The linear ozone model coefficients are formulated in terms of mixing ratio, but our analysis of photochemical adjustment is preferred in number density so that column ozone perturbations can be integrated by eye.

- l. 207: "quasi-steady state"? is this assumption justified? Our analysis applies to steady perturbations, which we now note explicitly: "*Both of these analyses consider the equilibrated response to steady perturbations.*" In other words, we are not claiming that the stratosphere is in a steady state; rather, we are restricting our analysis to steady perturbations.

- l. 217: does this mean that you investigate only tropical ozone here? Yes, this particular section in which we quantify photochemical adjustment considers the tropics, although the analysis with the linear ozone model coefficients could be performed at other latitudes. The subsequent analysis of photochemical adjustment in the Chapman+2 model is best restricted to the tropics, as is now noted.

- l. "uniform reduction" is not what is observed in the real atmosphere regarding tropical ozone depletion (WMO, 2022). Agreed, we now state explicitly: "*These perturbations are represented crudely as constant (or piecewise-constant), thereby deviating from realistic versions of these perturbations in order to highlight how photochemical adjustment can induce nontrivial vertical structure, even in response to uniform perturbations.*

- l. 229: How is G0 defined? How is this quantity derived? Also what altitude is z1? Why has a value of 60 km been chosen? Our analysis here is linear in the magnitude of $G_0$, which we define when describing Figure 4 as: "*The response to uniform ozone depletion below 60 km of $G_0 = 10^8$ molec $cm^{-3}$ is shown in Fig. 4a, with the prescribed depletion shown in blue.*" z1 (the top of the Cariolle v2.9 linear ozone model coefficients) is 80 km (now noted when discussing cooling). The

altitude of 60 km is chosen partially arbitrarily to fully encompass the stratosphere (the stratopause being around 50 km), and our key results are not unduly sensitive to this choice of altitude.

- l 251: G0 could be defined as a perturbation in mixing ratio—would this not be better? At least it should be considered, what a constant value of $G_0 = 10^8$ molec cm$^{-3}$ in ozone density means for mixing ratio. Ultimately, these simple perturbations are chosen for their illustrative power of showing photochemical adjustment. At the reviewer's recommendation, we consider below a constant mixing ratio perturbation of 0.1 ppb reduction. This perturbation is found to be less effective at illustrating photochemical adjustment, as it grows larger and larger towards the surface (perhaps following this line further would demand a tropopause cut-off), and the photochemical adjustment is harder to discern. Of course, there is still photochemical destabilization above 40 km. Figure 3 of this review shows this case.

[Figure]

Figure 3: Panel a: Constant mixing ratio perturbation (instead of constant number density perturbation as in the main body of the manuscript).

- l. 268: why only half? Photochemical adjustment gives you the sign of the change, but why is it half? That photochemical adjustment halves the response to the initial perturbation is a quantitative result from our calculation, not an input.

- l. 273: "entire stratosphere"? also when transport is important? Yes, this calculation is throughout the stratosphere, under the assumption of fixed transport tendencies.

- l. w89: is there a latitude range for the validity of the "40 km" statements? The latitudinal structure of the transition from destabilization to stabilization can be seen in Figure 3, where the transition is always above 40 km, ranging from around 40 km in the tropics to roughly 50 km near the pole.

- l. 297: I do not agree with this assessment if the Chapman cycle. See for example Fig. 5c in this manuscript. How does the comparison (Chapman vs. real world) look for ozone at other latitudes and seasons? Our analysis is now performed using

the Chapman+2 model. Interestingly, this statement is justified when considering the sign of the photochemical sensitivity, which is reproduced in both the Chapman Cycle and in the Chapman+2 model.

- l 333: argument for "photochemical equilibrium"? We have chosen to analyze photochemical adjustment in steady state, which we now state explicitly, as well as explicitly accounting for the effects of transport in steady state. We do not wish to claim that the stratosphere is, in fact, in steady state.

- l.334: the atmosphere is not isothermal? why is this assumption necessary? It only affects the chemical rate constants? is this correct? The assumption of an isothermal atmosphere only affects the rate constants, the vertical structure of which is not essential to reproducing photochemical adjustment. Assuming an isothermal atmosphere simplifies the analysis by removing an extraneous source of vertical structure that is not necessary to lead to the nontrivial structure emerging from photochemical adjustment itself.

- l. 343: You could use a compilation of photochemical parameters (Burkholder et al., 2019) instead of a textbook. No major impact for the questions treated in the manuscript. We agree, although we see no stylistic penalty for drawing our photochemical parameters from the textbook, which were drawn in turn from such compilations.

- l. 355: photolysis of ozone (R3) is seen here (and elsewhere in the manuscript) as a sink of ozone. This is not the case, if the $Ox = O + O3$ family is considered (independent of Chapman). This is explained below in terms of a null cycle for ozone (l. 356, l. 371), which reiterates many of the ideas of Ox; the loss of Ox is through R4. It is not clear to me why the Ox is not used, but then reintroduced (it seems) through the ozone null cycle. As noted above, we have updated our discussion of the photolytic sink of $O_3$, which remains a key concept in this manuscript, but is now discussed in terms of the odd oxygen family.

- l. 385: try ( for larger brackets Done.

- l. 387: an equation can not be negative (only a term in an equation) Fixed: "The photochemical sensitivities in equations X and X are strictly negative"

- l. 396: how is the situation for other months? The situation in other months supports our analysis, but is beyond the scope of this paper. The equinoctial month of September is quite similar to March, and there is a seasonal cycle over the poles in other months that is not considered here, although can be predicted from the solar zenith angle dependence in Figure 3.

- Eqs. 14 and 15: are these equations valid at the pole? The photolytic sensitivities are generic, with the latitude entering through the solar zenith angle, although neglecting higher-order corrections for sphericity (e.g., Prather and Hsu, 2019).

- l. 426: is "above 40 km" valid for all altitudes? Clarified for this section of the manuscript: "*As for the Cariolle v2.9 linear ozone model, we calculate the photochemical adjustment of the Chapman+2 model in the tropical stratosphere in response to ozone depletion and cooling.*"

- l. 429: why is it not possible to calculate P and L in a chemical model? These rates can in principle be calculated in the underlying model (MOBIDIC or UCI CTM), but they are not separately encoded in the linear ozone model coefficients, which is what we analyze.

- l. 451: suggest separating transport and catalytic chemistry? these two processes can be rather different. Our consideration of transport and catalytic chemistry is now using the Chapman+2 model.

- l. 456: you say "tropics" here — is the analysis also valid outside the tropics. This section is global, so we have re-worded for emphasis: "*The linear ozone models and Chapman+2 model robustly agree on the altitude of the photochemical regime transition from destabilization aloft to stabilization below. Thus, this regime transition is likely explained by simple physics that is shared among these models.*"

- l. 458: photolysis of ozone mostly drives the null cycle; see above. Reworded with same scientific message: "*Indeed, we find that the regime transition is controlled by which parts of the absorption spectrum for $O_3$ and $O_2$ have been attenuated versus which parts of the absorption spectrum are still active.*"

- l. 465: here (and elsewhere in the manuscript): what is meant by "column ozone"? I assume, partial column ozone above the level is question. And not total column ozone. I suggest being a bit more precise here. All usage of "column ozone" has been clarified, where intended usage of "overhead column ozone" has been revised as such.

- l. 482: "terminates"— coming from above of from below? Clarified wording: "*The bottom of the destabilization layer occurs robustly around 40 km because that is where overhead column ozone surpasses the stabilization threshold of $10^{18}$ molec cm$^{-2}$.*"

- l. 493: "approximating". Is this true for all conditions discussed here? E.g. at all seasons? I am also not sure that the approximation is needed. We approximate the solar zenith angle with the latitude for this global calculation. In general, the solar zenith angle will have diurnal and seasonal variability, the former being neglected without correction and the latter being neglected but with justification by restricting our focus to an equinoctial month. During the equinox, the solar zenith angle at solar noon equals the latitude.

- l. 554/555: this text is similar to the text in lns. 498/499. Thank you for pointing out this redundant language, the second instance of which has been removed.

- l. 581: "all the way to the surface"— isn't this extrapolation a bit dangerous given the fact that tropospheric chemistry is neglected? Reworded so as not to invoke caveats from tropospheric chemistry, which are not the main focus of this point: "*If ozone increased continuously, then there must have been an era when total column ozone was approximately $10^{18}$ molec cm$^{-2}$, during which our theory predicts that the ozone layer could have been completely photochemically destabilizing.*"

- l. 586: Do you have a citation for "surprising"? Replaced "surprising" with "counterintuitive", which we believe is self-evident. It is counterintuitive for an ODS to increase ozone given classification as ozone-depleting substance.

- l. 595: I am confused here. What is the major difference between the chemistry of MOBIDIC and the Cariolle scheme? Is there a chemical difference? Or is it mainly transport? Clarified elsewhere in the manuscript, which here now states: "*Photochemical sensitivities were quantified in two linear ozone models: Cariolle v2.9 and LINOZ. Photochemical adjustment was further quantified in column calculations using Cariolle v2.9 of the response to ozone depletion and stratospheric cooling.*"

- l. 639: incomplete citation Corrected to "conference paper" citation.

- l. 614: You say "tropics" here: is the paper on tropical ozone? That statement is true of the tropics, although we have clarified the geographic scope of the results elsewhere.

- l. 619: "down to the surface": this sounds very speculative; the analysis presented here is not for tropospheric altitudes in many respects. Agreed, reworded as follows: "*Paleoclimatic ozone layers with total column ozone below the destabilization threshold could have been completely destabilizing, amplifying ozone*

*perturbations and increasing the variability of UV light experienced by early life.*"

- l. 620: The MOBIDIC code and the employed coefficients of the Cariolle scheme (Eq. 1) should also be available. It would not be appropriate for us to publish this dataset, which we did not create and that is not (to our knowledge) elsewhere publicly available. Motivated readers will note that these coefficients were shared by personal communication with Daniel Cariolle (as noted in Section 2 and the Acknowledgments of the revised manuscript), to whom we are grateful.

- l 645: which journal? Journal added.

- l. 705: citation should contain pages or an electronic id. In the course of revisions, this citation is no longer needed.

- l. 709: is this citation correct? Do you mean WMO (2018)? Good catch; corrected.

We thank the reviewer again for their careful and constructive criticisms, the major revisions in accommodation of which we believe have improved this manuscript.

**References**

Bates, D. R., and M. Nicolet, 1950: The photochemistry of atmospheric water vapor. *Journal of Geophysical Research (1896-1977)*, **55 (3)**, 301–327, doi:10.1029/JZ055i003p00301.

Brasseur, G. P., and S. Solomon, 2005: *Aeronomy of the Middle Atmosphere: Chemistry and Physics of the Stratosphere and Mesosphere.* Springer, Dordrecht, Netherlands.

Cariolle, D., and M. Déqué, 1986: Southern hemisphere medium-scale waves and total ozone disturbances in a spectral general circulation model. *Journal of Geophysical Research: Atmospheres*, **91 (D10)**, 10 825–10 846, doi:10.1029/JD091ID10P10825.

Cariolle, D., and H. Teyssèdre, 2007: A revised linear ozone photochemistry parameterization for use in transport and general circulation models: Multi-annual simulations. *Atmospheric Chemistry and Physics*, **7 (9)**, 2183–2196, doi:10.5194/ACP-7-2183-2007.

Chapman, S., 1930: A theory of upper atmospheric ozone. *Memoirs of the Royal Meteorological Society*, **III (26)**, 103–125.

Crutzen, P. J., 1970: The influence of nitrogen oxides on the atmospheric ozone content. *Quarterly Journal of the Royal Meteorological Society*, **96 (408)**, 320–325, doi:10.1002/qj.49709640815.

Déqué, M., C. Dreveton, A. Braun, and D. Cariolle, 1994: The ARPEGE/IFS atmosphere model: A contribution to the French community climate modelling. *Climate Dynamics*, **10 (4)**, 249–266, doi:10.1007/BF00208992.

Jacob, D., 1999: *Introduction to Atmospheric Chemistry*. Princeton University Press.

Match, A., E. P. Gerber, and S. Fueglistaler, 2024: Protection without poison: Why tropical ozone maximizes in the interior of the atmosphere. *EGUsphere*, 1–29, doi:10.5194/egusphere-2024-1552.

McLinden, C. A., S. C. Olsen, B. Hannegan, O. Wild, M. J. Prather, and J. Sundet, 2000: Stratospheric ozone in 3-D models: A simple chemistry and the cross-tropopause flux. *Journal of Geophysical Research: Atmospheres*, **105 (D11)**, 14 653–14 665, doi:10.1029/2000JD900124.

Prather, M. J., and J. C. Hsu, 2019: A round Earth for climate models. *Proceedings of the National Academy of Sciences*, **116 (39)**, 19 330–19 335, doi:10.1073/pnas.1908198116.

Solomon, S., R. R. Garcia, and F. Stordal, 1985: Transport processes and ozone perturbations. *Journal of Geophysical Research: Atmospheres*, **90 (D7)**, 12 981–12 989, doi:10.1029/JD090ID07P12981.

---

## Referee Report (RR1)

*Second review of*

**"Beyond self-healing: Stabilizing and destabilizing photochemical adjustment of the ozone layer "**

*by A. Match et al.*

**General**

I see that the second version of the paper has seen a lot of work by the authors. I like in particular the change in the discussion in the paper throughout from the classical Chapman model to the Chapman+2 model. However, rereading the paper, I still feel that there are changes to the paper that did improve the readability and accuracy of the manuscript.

Therefore, I suggest that the authors consider the comments below and try to improve the paper further. I consider these changes 'minor'. If considered necessary, I would read the revised version of the paper again, but this should be the decision of the editor.

**Comments**

**Range of applicability**

I still think the paper could be clearer to where the ideas put forward here could be applied. For example, looking at Fig. 4 (and reading the caption) the proposed concept looks like a global (albeit 1D) result. However, in l. 176 it is stated that the calculation (and thus Fig. 4) is for tropical ozone. Furthermore, in the caption of Fig. 4 (and elsewhere in the paper) the 40 km demarcation is mentioned. From the explanation (which starts on page 8) I understand that 40 km is valid for the tropics (l. 176). If I am incorrect then the discussion following Fig. 4 should explain why the 40 km can be considered a global value (possibly one could repeat the calculation shown in Fig. 4 for a mid-latitude ozone profile). If on the other hand 40 km is more a tropical value, this should be clear throughout the paper (in particular in the conclusions).

**Equations 4 and 5**

I suggest formulating the assumptions used for deriving equations 4 and 5 more clearly. You assume

$$\frac{d\text{O}}{dt} = \frac{d\text{O}_3}{dt} = 0 \tag{1}$$

i.e., steady state between O and $\text{O}_3$ (I would not call this "typical equilibria"). Then you use reactions (R1) to (R6) to derive algebraic equations and then you replace the $\text{O}_2$ concentration by a constant value. Correct? Assuming constant $\text{O}_2$ concentration in (R1) would not give the desired result, would it?

**Model description and documentation**

I said in my first review: "..., MOBIDIC and the Cariolle scheme have certainly evolved over time during the decades since the cited reference in 1985. It would be good to have at least some information on the new parameters of the Cariolle scheme ...".

I appreciate the statement by the authors that the communication of the parameters of the scheme are a private communication and that this is a reason for not publishing these parameters. However, I suggest that the authors obtain the permission from D. Cariolle to make the parameters available to the public (in an appendix or in a table). I believe this would be of advantage to everybody considering to use the Cariolle scheme (and it would also be good for the paper).

**Some minor issues**

- l. 7: "that" $\longrightarrow$ "the enhanced"

- l. 13: "if" $\longrightarrow$ "when"

- l. 14: "where" $\longrightarrow$ "when"

- l. 20: "continual" $\longrightarrow$ "continuous"; but perhaps this is not the best way of describing the balance by $P$ and $L$ in the ozone layer

- 23: "reduce ozone at a particular altitude"

- l. 24: replace "locations" by "altitudes"?

- l. 24: do you also want to include Hartmann (1978) in this list of references?

- l. 26: you mention ODSs here, while Fig. 1 says CFCs. Perhaps make clear in the text that the same thing is meant.

- l. 37: "because" $\longrightarrow$ "from"

- l. 37: "allows" $\longrightarrow$ "allowing"

- l. 193: The top of the atmosphere (considered here) is 60 km – this is how I read these lines. Could you explicitly state this? And the numbering is from the top downwards – so the top of the atmosphere is $z_0$. Is this correct? I think the paper could be a bit clearer here.

- l. 193: Do you mean "$z_i$, where $i = 0, 1 - N$"? Or "$i = 1, N$"? Suggest to be accurate here. Also why not state what N is?

- l. 194: It would be good to give the value of the thickness employed here.

- l. 268: What is meant with "foundational" here? Drop this word here?

- l. 270: "rich theories" is not really clear here

- l. 285: "This assumption" is "$C_{O_2} = 0.21$" – correct?. But I think the central assumption is the equilibrium between O and $O_3$ (see also above).

- Eq. 17: try "\left(" and "\right)" instead of the brackets in the LaTeX equation.

- l. 583: lower than what?

---

## Author Response (AR3)

**Second response to Reviewers: "Beyond self-healing: Stabilizing and destabilizing photochemical adjustment of the ozone layer" (EGUSPHERE-2024-147)**

Aaron Match, Edwin P. Gerber, and Stephan Fueglistaler

August 1, 2024

We are pleased to hear that the reviewer supports our revisions to accommodate their thoughtful comments from the first round, and we are pleased to revise the paper to address their thoughtful minor recommendations on the second submitted version of the manuscript. Throughout this Response to Reviewers, reviewer comments will be in black and our author comments will be in blue.

**General**

I see that the second version of the paper has seen a lot of work by the authors. I like in particular the change in the discussion in the paper throughout from the classical Chapman model to the Chapman+2 model. However, rereading the paper, I still feel that there are changes to the paper that did improve the readability and accuracy of the manuscript.

Therefore, I suggest that the authors consider the comments below and try to improve the paper further. I consider these changes 'minor'. If considered necessary, I would read the revised version of the paper again, but this should be the decision of the editor.

We are pleased to hear that the reviewer approves of the Chapman+2 model for the development of a theory of photochemical adjustment. We thank them again for their comments that helped move us in this direction.

**Comments**

**Range of applicability**

I still think the paper could be clearer to where the ideas put forward here could be applied. For example, looking at Fig. 4 (and reading the caption) the proposed concept looks like a global (albeit 1D) result. However, in l. 176 it is stated that the calculation (and thus Fig. 4) is for tropical ozone. Furthermore, in the caption of Fig. 4 (and elsewhere in the paper) the 40 km demarcation is mentioned. From the explanation (which starts on page 8) I understand that 40 km is valid for the tropics (l. 176). If I am incorrect then the discussion following Fig. 4 should explain why the 40 km can be considered a global value (possibly one could repeat the calculation shown in Fig. 4 for a mid-latitude ozone profile). If on the other hand 40 km is more a tropical value, this should be clear throughout the paper (in particular in the conclusions).

Good point about the opportunity for more clarity in the caption about the geographic location of Figure 4. Given that this figure is for the tropics, in particular using the coefficients from the equator, we now label the caption as the "Equatorial ozone response to..." using the "equatorial coefficients from the Cariolle v2.9 linear ozone model" [italics added here for emphasis]. Note that the coefficients reasonable coherent throughout the tropics, so the equatorial column can be thought of as representing the entire tropical atmosphere.

We have further specified the location of applicability for our Chapman calculation, where overhead solar zenith angle is used to specify a tropical column as: "photochemical adjustment is destabilizing above 40 km in the tropical atmosphere" [emphasis added]

At line 455, we now state: "The bottom of the destabilization layer occurs robustly around 40 km *in the tropics*" [emphasis added]

In our Conclusions (line 595): "We have also characterized a new and unconventional region of photochemical destabilization above 40 km in the tropical stratosphere and even farther aloft at high latitudes."

**Equations 4 and 5**

I suggest formulating the assumptions used for deriving equations 4 and 5 more clearly. You assume

$$\frac{d\mathcal{O}}{dt} = \frac{d\mathcal{O}_3}{dt} = 0 \tag{1}$$

i.e., steady state between O and O3 (I would not call this "typical equilibria"). Then you use reactions (R1) to (R6) to derive algebraic equations and then you replace the O2 concentration by a constant value. Correct? Assuming constant O2 concentration in (R1) would not give the desired result, would it?

We have clarified this derivation in several ways. We now say: "For steady state solutions to the Chapman Cycle, the molar fraction of  $O_2$  is several orders of magnitude larger than that of O and O3, and will be treated as constant ( $C_{O_2} = 0.21$ ). Then, setting  $\partial O/\partial t = \partial [O_3]/\partial t = 0$ , these reactions can be algebraically solved to yield a quadratic equation for O3 at a given altitude:" We refer the curious reader to more details about this photochemical system to a related paper: "This photochemical system has been described in more detail in Match et al. (2024), which explains why the number density of ozone has an interior maximum in the tropical atmosphere." Note that we do in fact assume constant  $[O_2]$  concentrations in R1, as is standard for solving the Chapman system, such that the amount of  $O_2$  is prescribed as fully external to the Chapman system.

**Model description and documentation**

I said in my first review: "... MOBIDIC and the Cariolle scheme have certainly evolved over time during the decades since the cited reference in 1985. It would be good to have at least some information on the new parameters of the Cariolle scheme ... ".

I appreciate the statement by the authors that the communication of the parameters of the scheme are a private communication and that this is a reason for not publishing these parameters. However, I suggest that the authors obtain the permission from D. Cariolle to make the parameters available to the public (in an appendix or in a table). I believe this would be of advantage to everybody considering to use the Cariolle scheme (and it would also be good for the paper).

We are sympathetic to the idea that all data should be available for others. We have reached out to Dr. Cariolle to share with him that there is broad community interest in his dataset and to encourage him to consider sharing the data more broadly and systematically through a long-term public repository. We believe this could be a valuable resource for the community, but given that the data is not our intellectual property, this is ultimately not our decision. We hope that in the meantime, interested readers will be able to follow our footsteps and request the coefficients through personal communication.

**Some minor issues**

- l. 7: "that"  $\rightarrow$  "the enhanced" Revised.
- l. 13: "if"  $\rightarrow$  "when" Revised.
- l. 14: "where"  $\rightarrow$  "when" Revised.
- 1. 20: "continual" → "continuous" but perhaps this is not the best way of describing the balance by P and L in the ozone layer Continual → continuous.
- 23: "reduce ozone at a particular altitude" We believe the reviewer wants to help us forestall confusion about why the ODS can strictly reduce ozone while also leading to increases. We emphasize that our statement about strict reductions is for the *local* response, as we have stated: "As was perhaps first noted by Johnston (1972), emitting an ozone-depleting substance whose local chemical effects strictly reduce ozone can nonetheless cause ozone to increase at certain locations (subsequent treatments include Dütsch, 1979; WMO, 1985; Solomon et al., 1985; Fomichev et al., 2007; Meul et al., 2014)." Our statement is correct.
- l. 24: replace "locations" by "altitudes"? Revised.

- 1. 24: do you also want to include Hartmann (1978) in this list of references? Added.
- 1. 26: you mention ODSs here, while Fig. 1 says CFCs. Perhaps make clear in the text that the same thing is meant. We now emphasize that the model experiment imposes halocarbons which includes the more-familiar CFCs.
- l. 37: "because"  $\rightarrow$  "from" Revised.
- l. 37: "allows"  $\rightarrow$  "allowing" Revised.
- 1. 193: The top of the atmosphere (considered here) is 60 km? this is how I read these lines. Could you explicitly state this? And the numbering is from the top downwards? so the top of the atmosphere is z0. Is this correct? I think the paper could be a bit clearer here. The top of our model atmosphere is above 60 km, and 60 km is the fifth model level down from the top, at which we begin imposing our perturbation. We further clarify a point that benefits clarification following discussions of our method by noting at the end of that paragraph: "In this way, our calculation is locally linear in the overhead column ozone perturbation, but the dependence on overhead column ozone means that the ozone response at any given altitude depends nonlocally on the changes aloft."
- 1. 193: Do you mean "zi, where i = 0,1?N"? Or "i = 1,N"? Suggest to be accurate here. Also why not state what N is? Revised. We now state, "Both of these offline calculations have been performed on the discrete grid of the Cariolle v2.9 linear ozone model, with levels  $z_i$ , i = 1, 2, ..., N (where N = 91) numbered from the top of the atmosphere downwards, each with a thickness of  $\Delta z_i$  ranging from >3 km above 55 km to less than 1 km below the ozone maximum around 26 km."
- 1. 194: It would be good to give the value of the thickness employed here. Revised in previous comment.
- 1. 268: What is meant with "foundational" here? Drop this word here? We would like to retain the word foundational meaning to reflect that the Chapman Cycle provides the foundation for subsequent ozone science. As noted in Brasseur (2020), the puzzle of higher ozone at high latitudes compared to low latitudes led some analysts to seek to explain the ozone layer as a consequence of the high-energy particles that lead to the auroras over the poles. Chapman was the first scientist to explain ozone formation in terms of UV photochemistry, a foundational advance for all subsequent work on the topic. (Only later was the puzzle of high latitude ozone resolved by the work Brewer and Dobson.)
- l. 270: "rich theories" is not really clear here Removed "rich".

- 1. 285: "This assumption" is "CO2 = 0.21" ? correct?. But I think the central assumption is the equilibrium between O and O3 (see also above). As noted previously in this response, we have revised the reflect this central assumption more clearly: "For steady state solutions to the Chapman Cycle, the molar fraction of O2 is several orders of magnitude larger than that of O and O3, and will be treated as constant ( $C_{O_2} = 0.21$ ). Then, setting  $\partial O/\partial t = \partial [O_3]/\partial t = 0$ , these reactions can be algebraically solved to yield a quadratic equation for O3 at a given altitude:"
- Eq. 17: try "\left" and "\right)" instead of the brackets in the LaTeX equation. Revised.
- 1. 583: lower than what? Reworded for clarity: "Ozone-depleting substances can lead to increases in ozone at some altitudes, known as self-healing, which have been argued to result from ultraviolet fluxes reaching deeper into the atmosphere."

As a minor note, we have identified a missing factor of 2 to both terms on the r.h.s of Equation 15, which has been corrected in the Equation and in our calculation of Figure 3c. Our main emphasis using Equation 15 was on its sign, which was strictly unaffected by the omitted factor, so our conclusions are unaffected.

**References**

- Brasseur, G., 2020: *The Ozone Layer: From Discovery to Recovery*. University of Chicago Press, Chicago, IL.
- Dütsch, H. U., 1979: The search for solar cycle-ozone relationships. Journal of Atmospheric and Terrestrial Physics, 41 (7-8), 771–785, doi:10.1016/0021-9169(79)90124-7.
- Fomichev, V. I., A. I. Jonsson, J. de Grandpré, S. R. Beagley, C. McLandress, K. Semeniuk, and T. G. Shepherd, 2007: Response of the Middle Atmosphere to CO2 Doubling: Results from the Canadian Middle Atmosphere Model. *Journal of Climate*, **20** (7), 1121–1144, doi:10.1175/JCLI4030.1.
- Johnston, H., 1972: The Concorde, Oxides of Nitrogen, and Stratospheric Ozone. *Search*, **3** (8), 276–282.
- Match, A., E. P. Gerber, and S. Fueglistaler, 2024: Protection without poison: Why tropical ozone maximizes in the interior of the atmosphere. *EGUsphere*, 1–29, doi:10. 5194/egusphere-2024-1552.
- Meul, S., U. Langematz, S. Oberländer, H. Garny, and P. Jöckel, 2014: Chemical contribution to future tropical ozone change in the lower stratosphere. Atmos. Chem. Phys, 14, 2959–2971, doi:10.5194/acp-14-2959-2014.

- Solomon, S., R. R. Garcia, and F. Stordal, 1985: Transport processes and ozone perturbations. Journal of Geophysical Research: Atmospheres, 90 (D7), 12981–12989, doi:10.1029/JD090ID07P12981.
- WMO, 1985: Atmospheric Ozone: Assessment of our Understanding of the Processes Controlling its Present Distribution and Change. Tech. rep.